# Natural genetic variation impacts expression levels of coding, non-coding, and antisense transcripts in fission yeast

Mathieu Clément-Ziza[1,2,‡], Francesc X Marsellach[3,‡], Sandra Codlin[3,‡], Manos A Papadakis[4], Susanne Reinhardt[1], María Rodríguez-López[3], Stuart Martin[3], Samuel Marguerat[3,†], Alexander Schmidt[5], Eunhye Lee[3], Christopher T Workman[4], Jürg Bähler[3,*] & Andreas Beyer[1,2,**]

## Abstract

Our current understanding of how natural genetic variation affects gene expression beyond well-annotated coding genes is still limited. The use of deep sequencing technologies for the study of expression quantitative trait loci (eQTLs) has the potential to close this gap. Here, we generated the first recombinant strain library for fission yeast and conducted an RNA-seq-based QTL study of the coding, non-coding, and antisense transcriptomes. We show that the frequency of distal effects (*trans*-eQTLs) greatly exceeds the number of local effects (*cis*-eQTLs) and that non-coding RNAs are as likely to be affected by eQTLs as protein-coding RNAs. We identified a genetic variation of *swc5* that modifies the levels of 871 RNAs, with effects on both sense and antisense transcription, and show that this effect most likely goes through a compromised deposition of the histone variant H2A.Z. The strains, methods, and datasets generated here provide a rich resource for future studies.

**Keywords** antisense transcription; histone variant; non-coding RNA; QTL; *Schizosaccharomyces pombe*

**Subject Categories** Chromatin, Epigenetics, Genomics & Functional Genomics; Transcription

**Mol Syst Biol. (2014) 10: 764**

## Introduction

Variation in gene expression, which in turn is often caused by natural genetic variation, is a major factor causing intra-species phenotypic differences. Hence, investigating the influence of genetic variation on gene expression has been the focus of intense research. The identification of expression quantitative trait loci (eQTLs), that is, genomic regions that are linked with the expression of a specific transcript, has primarily been conducted using DNA microarrays (Jansen & Nap, 2001; Brem *et al*, 2002).

The advent of next-generation sequencing technologies is drastically changing the way we study the relationship between genotype and gene expression. High-throughput sequencing of cDNA (RNA-seq) has great potential to provide qualitatively new insights beyond mere mRNA quantification (Wang *et al*, 2009). RNA-seq is independent of gene annotation and provides information on all types of transcripts (including coding, non-coding, and antisense) and on the corresponding genotypes. However, many of these aspects have only been partially exploited. Previous RNA-seq-based eQTL studies have focused on measuring 'new' traits, giving rise to splicing QTLs (Lalonde *et al*, 2011; Battle *et al*, 2013), isoform-specific QTLs (Montgomery *et al*, 2010), and allele-specific eQTLs (Montgomery *et al*, 2010, 2011; Pickrell *et al*, 2010; Sun, 2012; Battle *et al*, 2013). Most of these studies have been limited to the detection of local eQTLs, so-called *cis*-eQTLs (Montgomery *et al*, 2010, 2011; Pickrell *et al*, 2010; Lalonde *et al*, 2011; Majewski & Pastinen, 2011), or have identified only a relatively small proportion of distant eQTLs (*trans*-eQTLs, Battle *et al*, 2013). *Cis*-eQTLs are located at or close to the genes whose expression they directly affect, while *trans*-eQTLs are remote from the genes whose expression they affect. It has been suggested that the vast majority of eQTLs act in *trans* (Brem *et al*, 2002; Yvert *et al*, 2003; Rockman & Kruglyak, 2006; Ackermann *et al*, 2013). Hence, to fully understand phenotypic variation, it will be necessary to design RNA-seq-based eQTL studies with the ability to detect all *trans*-eQTLs. Further, although the sequence variation

1 Biotechnology Centre, Technische Universität Dresden, Dresden, Germany
2 Cologne Cluster of Excellence in Cellular Stress Responses in Aging-associated Diseases (CECAD), University of Cologne, Cologne, Germany
3 Department of Genetics, Evolution & Environment and UCL Genetics Institute, University College London, London, UK
4 Center for Biological Sequence Analysis, Department of Systems Biology, Technical University of Denmark, Lyngby, Denmark
5 Biozentrum, University of Basel, Basel, Switzerland
 *Corresponding author. Tel: +44 203 108 1602; E-mail: j.bahler@ucl.ac.uk
 **Corresponding author. Tel: +49 221 478 84 429; E-mail: andreas.beyer@uni-koeln.de
 ‡These authors contributed equally to this work
 †Present address: MRC Clinical Sciences Centre, Imperial College London, London, UK

information contained in the RNA-seq data has been exploited for SNP detection (e.g. Piskol *et al*, 2013; Quinn *et al*, 2013), it has not been used in the framework of eQTL mapping, possibly due to the complexity of the studied organisms (Montgomery *et al*, 2010, 2011; Pickrell *et al*, 2010; Keane *et al*, 2011).

Here, we have conducted an expression QTL study characterized by a design enabling a high statistical power for association detection and by a broad investigation of pervasive expression beyond well-annotated coding genes (i.e. non-coding and antisense transcripts). The high statistical power contributed to the improved discovery of *trans*-eQTLs (thereby enabling the reliable detection of more than 2,000 *trans*-eQTLs), suggesting that previous studies may have been overestimating the fraction of *cis*-eQTLs. First, we generated a recombinant strain library for fission yeast (*Schizosaccharomyces pombe*) suitable for powerful QTL studies, which was subsequently subjected to high-resolution measurements of growth kinetics and strand-specific RNA-seq. Whereas microarray probes rely on a fixed 'reference genome', RNA-seq allows for the individualized quantification of transcripts taking genomic variation into account. We show that our approach, which explicitly includes individual genomes, reduces the potential for false-positive eQTLs. Further, because RNA-seq measures the actual transcript sequences of a given strain, it can also be used for genotyping the strain library. We developed a computational framework for the robust genotyping of recombinant strains using RNA-seq data, which eliminates the need for separate genotyping experiments. Finally, RNA-seq makes no assumptions about the structure of genomic features. In the context of QTL studies, it can thus be used to identify genetic variants affecting non-annotated features. Here, we present a striking example of a variation affecting antisense transcription of hundreds of *S. pombe* genes detected in this study.

# Results

## Generation of a recombinant *Schizosaccharomyces pombe* strain library

To generate the first fission yeast strain library suitable for QTL studies, we selected an independent isolate that is not derived from Leupold's widely used 968 $h^{90}$ strain (968; standard laboratory strain, Leupold, 1950). In order to enable the detection of association at a high statistical power, we selected closely related parental strains to reduce the genetic complexity of the library (Brown *et al*, 2011). Reproductive barriers have been reported between *S. pombe* wild isolates (Kondrat'eva & Naumov, 2001), which could lead to low recombination and biased segregation of polymorphic markers (Chambers *et al*, 1996). To avoid such problems, we characterized the mating and segregation of polymorphic markers among fission yeast wild isolates (D. Jeffares *et al*, in preparation) available in strain collections (Brown *et al*, 2011). Based on these analyses, we chose Y0036 (isolated in South Africa) as a suitable mating partner for the 968 strain to produce a QTL library. Crosses between the Y0036 and 968 strains showed 55% spore survival by tetrad analysis. Moreover, microsatellite markers (Patch & Aves, 2007) were polymorphic between these two strains, and Affymetrix-based genotyping showed no segregation biases, with the exception of a low recombination rate on a large region of Chromosome I. This result

was later confirmed with the RNA-seq genotyping of the QTL library (see below).

The strain library was constructed with a customized protocol to ensure that matings occurred exclusively between the Y0036 and 968 strains, but not within parental strains (Materials and Methods). This approach also showed that increasing the number of crosses, that is, generating F2 segregants, significantly helped to improve the resolution of the QTL library (F2 segregants contained more recombinations than in F1 segregants, $P = 0.0111$, one-sided Wilcoxon's rank-sum test), as previously shown (Parts *et al*, 2011; Liti & Louis, 2012). In total, we generated 44 recombinant segregant strains from the cross between Y0036 and 968 (Supplementary Table S1). Note that fission yeast strains normally grow as haploid cells, and all our segregants were haploid.

## Phenotyping of strain library by high-resolution growth measurements and RNA-seq

We analyzed the growth of the recombinant strains in liquid media in standard conditions using the BioLector instrument (Kensy *et al*, 2009). This approach allowed us to acquire real-time cellular growth kinetics and high-resolution growth curves, which in turn facilitated the resolution of individual growth variables with high precision for the parental and all segregant strains of the library (Supplementary Fig S1A, Supplementary Dataset S1). We extracted from these data a number of growth parameters, including the lag phase (response time), the maximum specific growth rate ($\mu_{max}$, slope in exponential phase of growth converted into population doubling time), the growth efficiency (gain in biomass provided by nutrients in medium), and the area under the growth curve (Supplementary Fig S1B–D, Supplementary Dataset S2). All growth traits showed variability within the library. Interestingly, the growth efficiency and maximum specific growth rate were higher in the parental strains than in 96.6 and 94.1% of the segregants, respectively (Supplementary Fig S2). This apparent outbreed depression suggests an adaptive evolution of the parental strains during their lineage that resulted in epistatic interactions between different loci (Edmands, 1999; Gimond *et al*, 2013).

We also determined the viability in stationary phase of the parental and segregant strains after 36 h of growth in liquid medium. A large fraction of the segregants (~46%) showed a high average population viability corresponding to 76–100% (Supplementary Fig S3, Supplementary Dataset S3). The viabilities of the parental strains were very different from each other: 38.0 and 87.6% for Y0036 and 968, respectively.

The library was also subjected to a molecular characterization of genotypes and expression phenotypes via transcriptome sequencing at high depth (50 base long strand-specific RNA-seq, average RNA-seq statistics: Table 1, detailed: Supplementary Table S2). We could reliably measure: (i) the expression of 6,464 previously identified transcripts (5,036 protein-coding and 1,428 non-coding transcripts, Supplementary Dataset S4, Supplementary Table S2), and (ii) the antisense expression at the loci of 4,133 coding genes (Supplementary Dataset S5, Supplementary Table S2) in 65 samples (including biological replicates) derived from 46 distinct strains (Supplementary Table S2). The average Pearson's product-moment correlation coefficient (Pearson's $r$) between the biological replicates was $r = 0.97$ (Supplementary Fig S4).

**Table 1.  RNA-seq average statistics.**

| | |
|---|---|
| Average number of reads | 22,372,046 |
| Average mapping efficiency | 45% |
| Average effective depth | 37.5× |

For detail see Supplementary Table S2.

## Genotyping of the strain library by RNA-seq

eQTL studies require both the genotypes and the expression profiles of each individual of the studied population. We developed a strategy enabling the genotyping of a recombinant strain library through RNA-seq (Fig 1). Thus, separate genotyping experiments of the segregant strains were not needed.

First, the genomes of the parental strains were sequenced at great depth (273× and 244× for 968 and Y0036, respectively), allowing us to detect potentially all genomic variants (Supplementary Fig S5). We identified 4,570 high-confidence genomic polymorphisms between the two parental strains. These genetic variants included 3,865 single-nucleotide polymorphisms (SNPs) and 705 small insertions and deletions (indels) (Supplementary Dataset S6), representing a minimum genomic divergence of 0.05% between the two parental strains. This degree of divergence is comparable to the average divergence between two humans (~0.1%; Jorde & Wooding, 2004), but it is less than the divergence typically observed between progenitors of model recombinant inbred populations; for instance, the widely used *S. cerevisiae* RMxBY strains are ten times more divergent (Brem *et al*, 2002; Bloom *et al*, 2013).

Subsequently, we used the RNA-seq data of the segregants to detect genomic variations and thus determine their haplotype, at 4,481 polymorphic sites, with 89 sites being excluded from this analysis (Fig 2, Supplementary Dataset S7, Materials and Methods). On average, about half of the sites (47.7%) could be directly genotyped with high certainty, which was sufficient to identify haplotype blocks and therefore infer genotypes at the remaining sites. Ambiguous sites (e.g. between two haplotype blocks) were not called. Overall, 97.3% of the polymorphic sites could be genotyped (Materials and Methods). From this genotype data, we inferred 765 recombination breakpoints (Supplementary Fig S6, Supplementary Dataset S8), that is, on average 17.4 recombination sites per segregant. A set of 708 genetic makers was then assembled (Materials and Methods, Supplementary Dataset S9) to perform the subsequent QTL analyses, with an average interval between markers of 17.8 kb.

We noticed that a large part of Chromosome I exhibited a reduced recombination frequency (Fig 2, Supplementary Fig S6). This region corresponds to a 2.23 Mbp pericentric inversion shared by a minority of *S. pombe* isolates (Brown *et al*, 2011), including the 968 strain, but not the other parental strain (Y0036). This inversion on Chromosome I lowered the frequencies of crossovers, which resulted in a reduced genetic resolution in this region (distances between informative genetic markers in this region were on average three times larger, Supplementary Information, Supplementary Figs S6 and S7).

## QTL detection

Next, we used the RNA-seq data from the 65 samples (derived from 44 distinct recombinant strains and the two parental strains, Supplementary Table S2) to determine the expression levels of 6,464 coding and non-coding transcripts (Supplementary Dataset S4). QTL mapping was done using a method that we previously developed (Michaelson *et al*, 2010). Because we noticed the presence of population structure in the cross, the QTL mapping method was improved to also account for population structure and missing genotype data (Materials and Methods). At an empirical false discovery rate (FDR) of 5%, we detected 2,346 eQTLs that were linked to 2,179 transcripts, 682 of which were non-coding (Fig 3B, Supplementary Datasets S10 and S11). Almost half of these eQTLs (1,040; 44%) had a FDR of < 1%, which underlines the high statistical power of this study. We mapped the variation of antisense expression levels of 4,133 coding genes (Supplementary Dataset S5) to genetic variation, resulting in an 'antisense eQTL' (aseQTL) study. At 5% FDR, we detected 2,066 aseQTLs affecting 1,911 antisense expression traits (Fig 3C, Supplementary Datasets S12 and S13). QTLs were also found for all the growth traits measured (at a FDR of 10%, Fig 3A, Supplementary Dataset S14).

## Accounting for individual genomes improves transcript quantification

In microarray-based expression studies, sequence variation in probe regions can affect the hybridization efficiency. Because this leads to an allele-specific signal bias, sequence variation can inflate the number of false *cis*-eQTL calls (Alberts *et al*, 2007; Benovoy *et al*, 2008; Verdugo *et al*, 2010). Notably, RNA-seq studies are neither immune to such artifacts (Degner *et al*, 2009) as transcript quantification usually involves the mapping of sequence reads to a reference genome. Reads from individuals carrying the allele from the reference genome are more likely to be mapped than reads from individuals carrying a polymorphic allele, because alignment algorithms treat sequence differences as mismatches. Such bias may lower the expression estimates of polymorphic alleles, which would bias eQTL calls in a similar way as in microarray studies. The effects of this potential artifact in whole-genome RNA-seq-based eQTL mapping have not been investigated to our knowledge, presumably because previous studies have used human samples where single-base genotype information is often not available (Montgomery *et al*, 2010, 2011; Pickrell *et al*, 2010; Lalonde *et al*, 2011; Majewski & Pastinen, 2011). However, such biases have already been noticed in the framework of allele-specific expression in heterozygous diploid individuals (Degner *et al*, 2009; Turro *et al*, 2011; Reddy *et al*, 2012; Satya *et al*, 2012; Franzén *et al*, 2013; Pandey *et al*, 2013; Stevenson *et al*, 2013). As a solution to this problem, it has been proposed to mask or exclude all or part of the polymorphic positions in the reference genome prior to read-mapping (Degner *et al*, 2009; Stevenson *et al*, 2013), or to use multi-mapping strategies (Turro *et al*, 2011; Satya *et al*, 2012).

In this study, haplotype information is available at single-base resolution from the DNA sequencing of the parental strains. Thus, we used our data to investigate potential biases caused by intragenic polymorphisms by comparing the gene expression quantification obtained from alignments to strain-specific genomes versus the reference genome (Supplementary Information, Supplementary Fig S8).

In order to initially explore the potential benefits of strain-specific genome mapping, we simulated the sequencing of the transcriptome of the two parental strains and of three segregants and compared the accuracy of the expression quantification obtained using strain-specific versus reference genome mapping.

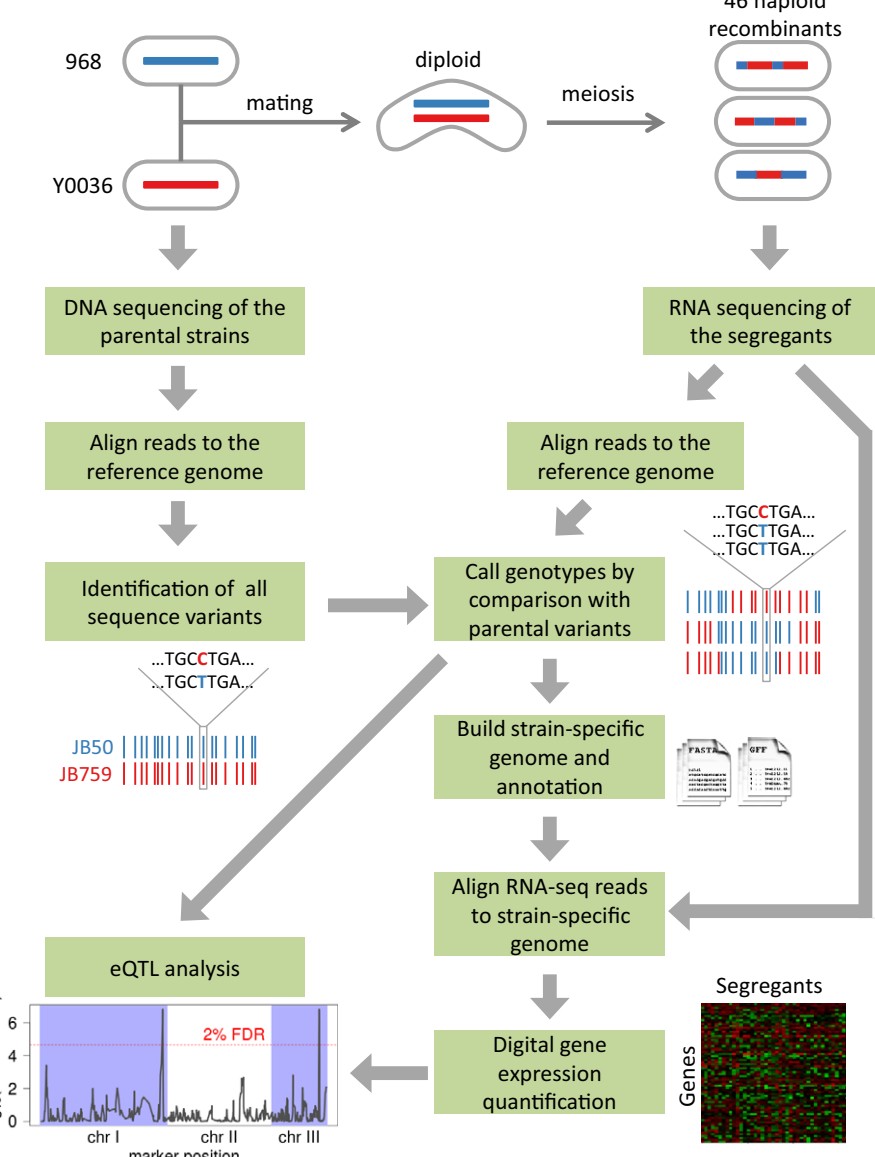

**Figure 1. Scheme of RNA-seq-based eQTL study.**

The *Schizosaccharomyces pombe* strain 986 (isogenic with standard laboratory strain) was crossed with an independent wild isolate (Y0036). Small genetic polymorphisms of the parental strains were almost exhaustively characterized through DNA sequencing. Segregant transcriptomes were quantified through RNA-seq. The RNA-seq data were also used for the detection of sequence variations in the transcribed regions of segregants. Genotypes and haplotypes of the segregants were reconstructed by comparing the identified genetic variants to the known variants of the parental strains (Materials and Methods). Subsequently, for each strain, we reconstructed its genome sequence based on the previously determined genotypes. The RNA-seq data were then re-mapped on these strain-specific genomes to avoid quantification biases due to sequence variations. From these alignments, we extracted digital gene expression profiles that were used as traits for eQTL mapping.

RNA-seq reads were simulated taking natural variation into account and in modeling sequencing errors. Considering all simulations together, we detected 1,644 (5.1%) differentially quantified expression traits. Moreover, as expected, the large majority of the measurements were increased by strain-specific genome mapping (98.3%, Supplementary Fig S9A). Next, we compared the simulated gene expression measurements to the input expression values. The expression obtained with strain-specific genome mapping was closer to the input value for 89.7% of the traits differentially quantified. Furthermore, the

measurement error (absolute difference between the log-transformed measured and input expression levels) was significantly lower when using strain-specific genome mapping ($P < 2.2$e-16, paired Kolmogorov–Smirnov test, Supplementary Fig S9B). Although significant, the magnitude of the gains of accuracy was small since the reduction of the measurement error exceeded 10% in only 2.2% of the cases.

Next, we tested the effect of strain-specific genome mapping on our real data (Supplementary Information, Supplementary Fig S8). Indeed, most genes containing a polymorphism (92.6%)

 

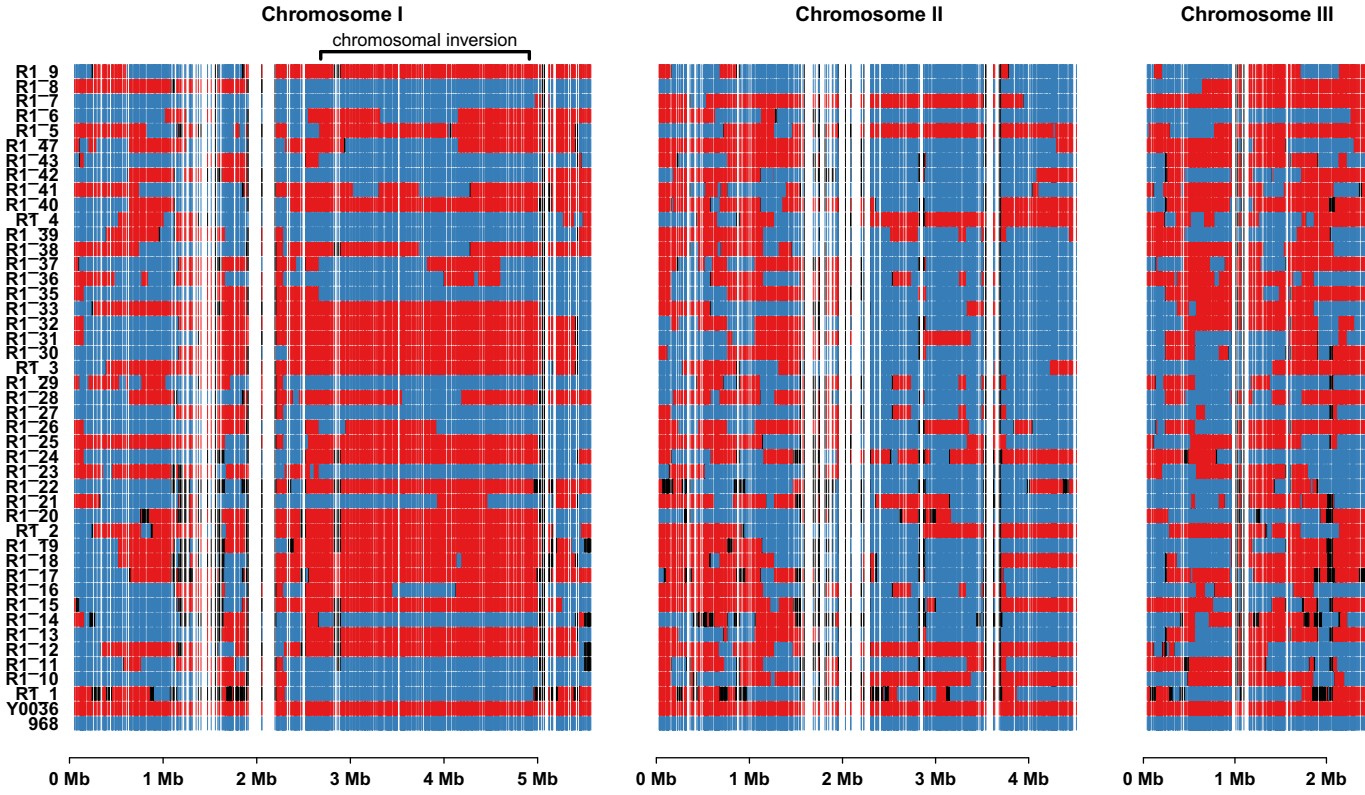

**Figure 2.  Genotype of strain library obtained through RNA-seq-based genotyping.**
Each row corresponds to one strain, with inherited markers shown across the three chromosomes. Blue bars are markers inherited from the 968 parent, red bars from the Y0036 parent, and black bars correspond to unknowns.

were differentially quantified in at least one sample (Fig 4A). Moreover, as expected and similarly to what was observed with simulated data, most of the affected measurements (98.3%, same fraction as in simulations above) were increased by strain-specific genome mapping (Fig 4B). Despite these artifacts, eQTL detection was only marginally affected, possibly due to the small genetic divergence between the progenitor strains (Discussion). Taken together, these results show that aligning RNA-seq data against individualized genomes marginally improves the transcript quantification, while ignoring individual sequence variation can inflate the number of falsely detected *cis*-eQTLs (Supplementary Information).

**trans-eQTLs greatly exceed cis-eQTLs in abundance**

It is generally assumed that *cis*-eQTLs can be detected more easily than *trans*-eQTLs (Ackermann *et al*, 2013; Schadt *et al*, 2003; Doss *et al*, 2005; Gerrits *et al*, 2009; Holloway *et al*, 2011): First, because direct effects on local genes are often stronger than indirect effects and second, for the statistical reason that searching for *trans*-eQTLs involves testing of a much larger number of hypotheses. Accordingly, *cis*-eQTLs typically make up a large fraction of all detected eQTLs: 15–50% for the microarray-based eQTL studies (Doss *et al*, 2005; Ronald *et al*, 2005; West *et al*, 2007; Breitling *et al*, 2008; Ghazalpour *et al*, 2008; Atwell *et al*, 2010) or even 97.5–100% for the existing RNA-seq-based eQTL studies (Montgomery *et al*, 2010,

2011; Pickrell *et al*, 2010; Sun, 2012; Battle *et al*, 2013). In contrast to these previous studies, we detected a lower fraction of *cis*-eQTLs (243 *cis*-eQTLs, 10.4% of all eQTLs, Fig 3B). This discrepancy with previous work can be explained by several factors, but we think that it results from the increased statistical power for detecting QTL, which may have arisen from the experimental design, and/or our experimental and analytic approaches (Discussion).

**Non-coding RNA expression is strongly affected by genetic variation**

RNA-seq enables the quantification of entire transcriptomes, including non-coding RNAs (ncRNAs) (Wilhelm *et al*, 2008; Wang *et al*, 2009; Costa *et al*, 2010; Marguerat & Bähler, 2010). We know little about the extent to which variation of ncRNA levels can be explained by natural genetic variation (Schadt *et al*, 2008; Montgomery *et al*, 2010; Pickrell *et al*, 2010; Kumar *et al*, 2013). The high sequencing depth used here and the extensive annotation of the fission yeast genome enabled us to quantify transcript levels for 1,428 annotated ncRNAs. Thus, this analysis presents the first comparative eQTL mapping for coding versus non-coding transcript levels at a genomic scale. We detected 753 significant (FDR < 0.05) eQTLs that together affected the expression of 682 distinct ncRNAs. Surprisingly, the fraction of ncRNAs targeted by at least one eQTL was even higher than the respective fraction of coding RNAs (47.8% versus 29.7%, respectively; enrichment for non-coding

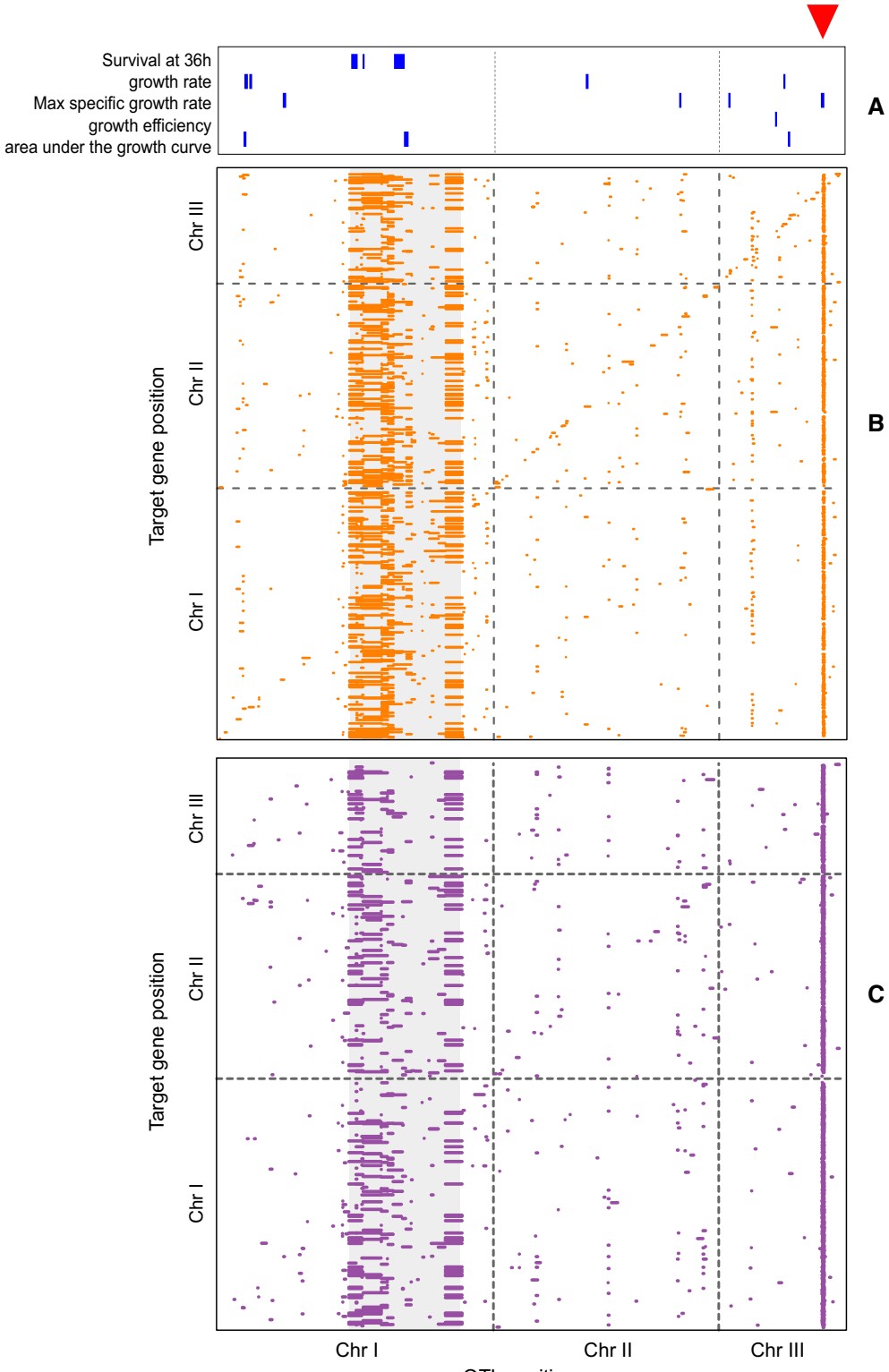

**Figure 3.   QTL maps for growth traits, sense expression, and antisense expression.**
The red triangle indicates the position of the Chromosome III hotspot (*swc5* locus).

A   Growth QTL (10% FDR) obtained from high-resolution growth phenotyping in liquid culture and the survival rate at 36 h.
B   eQTL map for 6,464 expression traits (all annotated genes, coding, and non-coding). Each dot represents a significant eQTL (5% FDR) where locus positions are represented on the *x*-axis and gene positions on the *y*-axis. The zone in gray corresponds to the Chromosome I inversion.
C   aseQTL map for 4,133 antisense expression traits.

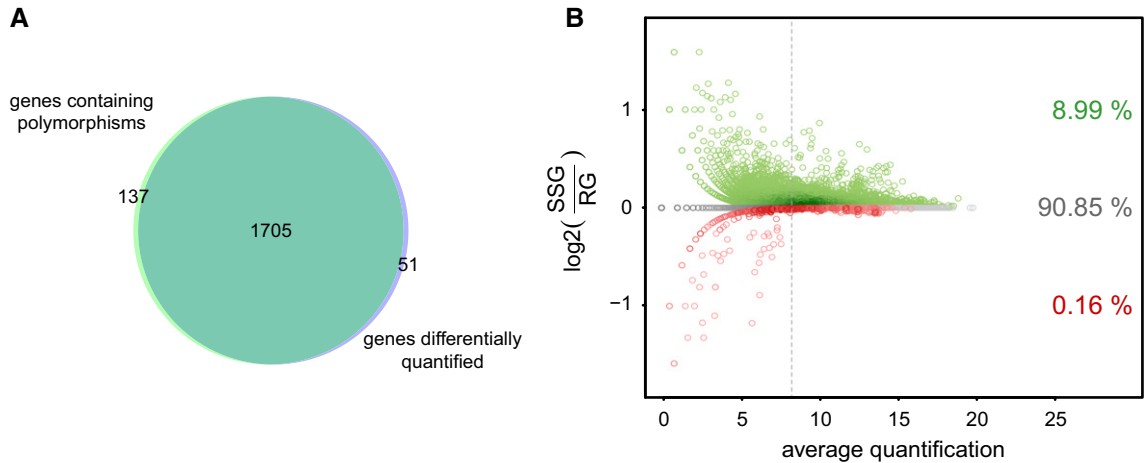

**Figure 4.  Accounting for individual genomes affects expression quantification.**

A  Venn diagram showing the overlap between genes with at least one polymorphism and genes differentially quantified in at least one sample when mapping against the strain-specific genome (SSG) as compared to mapping against the reference genome (RG).

B  Comparison of expression quantification measurements obtained by the SSG and RG mapping strategies. Green (red): measurements quantified higher (lower) using SSG versus RG. Color intensities reflect the density of points. The vertical line denotes the median gene expression.

genes among eQTL targets $P < 2.2e\text{-}16$, one-sided Fisher's exact test). To rule out that only a small number of loci affecting many ncRNAs are responsible for this trend, we re-computed these fractions after excluding all eQTL hotspots (see below) and the Chromosome I inversion. The trend remained the same, although the difference was not statistically significant anymore (7.2% versus 6.3%, respectively, $P = 0.14$, one-sided Fisher's exact test). Finally, we noticed that the ncRNAs were significantly more often affected by *cis*-eQTLs than coding RNAs (3.2 and 5.7% *cis*-eQTL for coding and non-coding RNAs, respectively, $P = 2.54e\text{-}05$, one-sided Fisher's exact test). Altogether, these findings indicate that the expression of non-coding RNAs is at least as much affected by genetic variation as the expression of protein-coding RNAs.

To further investigate the importance of non-coding RNAs as effectors of eQTLs, we predicted the most likely causal gene for each eQTL (Supplementary Dataset S11). We noticed that ncRNAs were enriched among the genes predicted as being most likely causal (31.2% of the predicted causal genes were ncRNAs, whereas 21.7% of all genes at eQTL are ncRNA; $P < 10^{-15}$, Fisher's exact test). This result suggests that non-coding RNAs substantially contribute as effectors of the genetic variation of gene expression. Note that we identified 12 *trans*-linkages that involved eQTL containing either no genes or only non-coding genes. Those cases (Supplementary Dataset S15) constitute a set of eQTL where the causal gene is most likely non-coding.

**eQTL and aseQTL hotspots**

Loci affecting gene expression in eQTL studies are non-randomly distributed in the genome: A few 'hotspot' loci can regulate the expression of numerous genes (Brem *et al*, 2002; Ghazalpour *et al*, 2008; Atwell *et al*, 2010). We identified 8 genomic regions significantly enriched for eQTLs (Fig 3B, Supplementary Fig S10A, Supplementary Table S3). These hotspots were robust with respect to different strategies for normalization of the expression measurements

and different FDR cutoffs, suggesting that they do not represent statistical artifacts (Supplementary Fig S10B; Wang *et al*, 2007; Williams, 2006). Moreover, the method that we used to assess the significance of QTL follows a previously suggested principle for preventing the detection of spurious QTL hotspots (Breitling *et al*, 2008). Four hotspots were located in the region of the Chromosome I inversion, and eQTLs in this region together targeted 44% of all transcripts affected by any eQTL (950 out of 2,179). Thus, this large inversion has a strong influence on the expression of many genes (Supplementary Information, Supplementary Fig S10A). Four out of eight eQTL hotspots were also aseQTL hotspots (Supplementary Fig S10A, Supplementary Table S3). Further, the eQTL hotspots 1, 2, 4, and 8 were also QTL for several growth traits (Fig 3). This result further suggests that the eQTL mapping and hotspot detection results are not statistical artifacts, since the growth traits represent independently measured phenotypes.

Functional enrichment analysis revealed hotspot-specific functions enriched among the target genes (Supplementary Tables S3 and S4). Targets of hotspot 1 were enriched for ribosome biogenesis (Supplementary Table S3). The same hotspot also affected a growth trait (growth rate). Ribosome biogenesis has been shown to play an important role in growth via the Tor pathway (Schmelzle & Hall, 2000; Mayer & Grumdt, 2006). Hotspot 2, linked to genes enriched for stress response (Supplementary Table S3), is also linked to survival in stationary phase. The importance of molecular stress response for aging and survival is well established in diverse species, including fission yeast (Sohal & Weindruch, 1996; Zuin *et al*, 2010). Taken together, these results highlight the utility of this fission yeast library for studying the relationships between molecular and macroscopic traits.

**A frameshift in *swc5* causes major eQTL hotspot**

Whereas several hypotheses could explain the strong impact of the inversion on Chromosome I on gene expression (Supplementary

Information), the cause of a second major hotspot on Chromosome III was not immediately apparent. This other eQTL hotspot (hotspot 8), influenced the expression of 871 genes (Supplementary Fig S10A, Supplementary Table S3). This hotspot was also the strongest aseQTL hotspot. It was linked to 1,384 antisense traits, that is, it is affecting antisense transcription even more than sense transcription. To our knowledge, hotspot 8 shows more widespread gene expression effects than any other hotspot reported so far (note that the effect of the Chromosome I inversion is spread over several loci). It has been discussed that eQTL hotspots, especially the strongest ones, can constitute artifacts (Williams, 2006; Wang *et al*, 2007; Breitling *et al*, 2008). Here, two lines of evidence suggest that hotspot 8 represents a true biological event: (i) This hotspot was also linked to cellular growth, that is, an independent and distinct trait (Fig 3); and (ii) it was robust to the application of different normalization strategies (Supplementary Fig S10B), which is notable as insufficient correction for confounding factors is believed to be a major source of false eQTL hotspots (Williams, 2006; Michaelson *et al*, 2010).

Because of its extraordinary strength, we wanted to unravel the molecular cause of hotspot 8. In previous eQTLs studies, the genetic variations explaining eQTL hotspots were often deletions of entire genes or mutations in the coding sequence of genes resulting in *de facto* knockouts (Brem & Kruglyak, 2005; Keurentjes *et al*, 2007; Breitling *et al*, 2008; Foss *et al*, 2011). Therefore, we focused our attention on non-silent polymorphisms within the hotspot region (Supplementary Fig S11). Six protein-coding genes in this region were affected, five of them through amino acid substitutions. The only exception was a frame-shift mutation in the Y0036 allele (p.Asn74Lysfs*2), potentially resulting in an early translation termination of the Swc5 protein (Swc5-fs; Fig 5A). The truncation of Swc5 in strains carrying the Y0036 allele (*swc5-Y0036*) was confirmed using targeted proteomics: Whereas we reliably detected peptides located before and after the frameshift in strains carrying the 968 allele (*swc5-968*), we only detected peptides located before, but not after the frameshift in *swc5-Y0036* strains (Fig 5A and B, Supplementary Fig S12). This result was observed for all analyzed samples and replicates (Supplementary Fig S12).

Previous work has shown that genes causal for eQTL hotspots often affect their own expression, resulting in *cis*-eQTLs (Loguercio *et al*, 2010). The *swc5* gene showed the strongest *cis*-eQTL in this region (FDR < 0.0017), further suggesting that *swc5* could be causal. To corroborate this hypothesis, we performed RNA-seq of a strain with a *swc5* deletion (Δ*swc5*). The correlation of expression between the Δ*swc5* and *swc5-Y0036* (*swc5-fs*) strains was greater than between the Δ*swc5* and *swc5-968* (wild-type *swc5*) strains (Fig 5C). Notably, significant differences in correlation were restricted to genes linked to the hotspot (expression of the hotspot target genes in Δ*swc5* was more strongly correlated with *swc5-fs* segregants than with *swc5*$^+$ segregants, $P = 1.3 \times 10^{-11}$, Wilcoxon's signed-rank test). Similar results were obtained with respect to antisense expression (Pearson's *r* correlation between Δ*swc5* replicates and *swc5-fs* segregants was greater than between Δ*swc5* replicates and *swc5*$^+$ segregants when considering the antisense traits linked to *swc5* locus, $P < 0.001$, one-sided Wilcoxon's rank-sum test). Moreover, clustering analysis confirmed that the

Δ*swc5* expression profile clusters with *swc5-Y0036* (Supplementary Fig S13) and genes being differentially expressed in the knockout versus wild-type strain overlapped significantly with hotspot 8 targets (sense expressions: $P < 10^{-58}$, antisense expression: $P < 10^{-11}$, one-sided Fisher's exact test, Supplementary Fig S14). We attribute the remaining trait variance that is not explained by the *swc5*-knockout to other variants in linkage disequilibrium with *swc5*, effects of auxotrophic markers in the knockout strain, or simply noise in the data. Taken together, these observations suggest the *swc5* frameshift (*swc5-fs*) as the major genetic cause underlying hotspot 8.

### *swc5-fs* reduces H2A.Z deposition

Swc5 is a component of the Swr1 protein complex controlling the chromosomal deposition of the histone variant H2A.Z (named Pht1 in fission yeast; Krogan *et al*, 2003; Mizuguchi *et al*, 2004; Kobor *et al*, 2004; Wu *et al*, 2005; Zofall *et al*, 2009; Buchanan *et al*, 2009). We therefore hypothesized that the strong phenotype of the *swc5-fs* allele is mediated by altered H2A.Z localization. In fission yeast, H2A.Z is involved in the regulation of antisense transcription (Zofall *et al*, 2009), maintenance of subtelomeric and centromeric gene silencing (Buchanan *et al*, 2009; Hou *et al*, 2010), chromosome segregation (Kim *et al*, 2009; Hou *et al*, 2010), and genome stability (Kim *et al*, 2009). To test the impact of *swc5-fs* on H2A.Z deposition, we performed genome-wide chromatin immunoprecipitation of H2A.Z coupled with deep sequencing (ChIP-seq) in the two parental strains and in Δ*swc5* strains. H2A.Z was enriched at the first nucleosome after the transcription start site (Fig 5D), as previously reported (Buchanan *et al*, 2009). Although the localization of H2A.Z was similar in the two parental strains, Y0036 exhibited a significant reduction of H2A.Z occupancy at the first nucleosome after the transcription start site (Fig 5D, 47.9%, $P < 10^{-15}$, one-sided Wilcoxon's signed-rank test). The same reduction of H2A.Z occupancy at the +1 nucleosome was also observed in the Δ*swc5* strains, thus confirming the specificity of the effect (Fig 5D, 53.2%, $P < 10^{-15}$, one-sided Wilcoxon's signed-rank test). We also observed marginal but significant differences between Y0036 and Δ*swc5* ($P = 5.83e-09$, Wilcoxon's signed-rank test, but small in magnitude, 5.4% on average), suggesting that the change in H2A.Z deposition in Y0036 is not exclusively caused by the *swc5* frameshift. The reduced H2A.Z occupancy with no modification of its localization observed in Δ*swc5* and *swc5-fs* strains is consistent with previous findings in *Saccharomyces cerevisiae* showing that the function of Swc5 in the SWR1 complex is related to the H2A–H2AZ histone exchange, but neither to the H2AZ nor the nucleosome binding (Wu *et al*, 2005). In addition, we noticed that in the 968 strain, H2A.Z occupancy was significantly greater at target genes of the *swc5*-eQTL as compared to non-target genes (Fig 5E, $P < 10^{-15}$, Wilcoxon's rank-sum test). Similar results were obtained for antisense expression: H2A.Z occupancy at *swc5* aseQTL targets was greater than at non-target genes (Supplementary Fig S15, $P < 10^{-15}$, one-sided Wilcoxon's rank-sum test). Thus, the *swc5*-eQTL and *swc5*-aseQTL coincides with stronger H2A.Z deposition under the wild-type condition (968 genetic background). These findings further corroborate the causal role of *swc5-fs* and the resulting changes in H2A.Z deposition for the eQTL hotspot on Chromosome III.

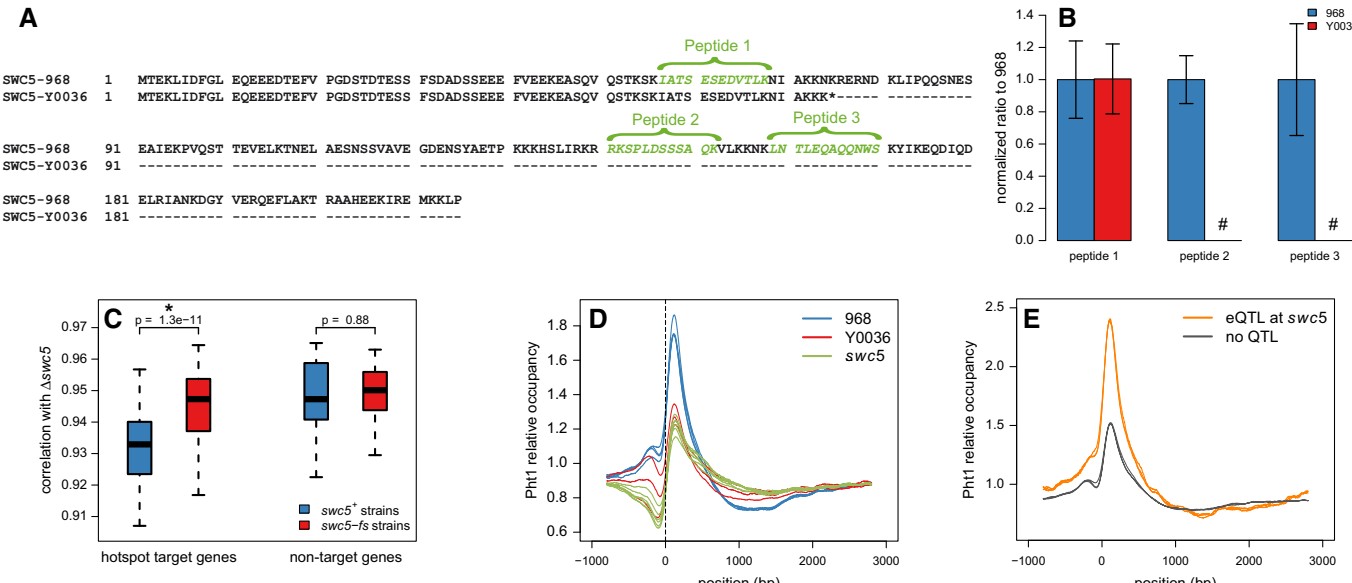

**Figure 5. swc5 frameshift as molecular basis of the Chromosome III hotspot.**

A  Sequence of Swc5 in the 968 and Y0036 genetic backgrounds, the latter containing the frameshift polymorphism *swc5-fs*.

B  Average ratios and standard deviations for all three peptides determined from three independent replicates of strains 968 and Y0036. MS signals with signal-to-noise ratios of < 3 are not considered (#). Error bars indicate standard deviation. The peptides located after the frameshift could not be detected.

C  Pearson's product-moment correlation coefficient of expression patterns between Δ*swc5* and *swc5-Y0036* (frameshift polymorphism; *swc5-fs*) and *swc5-968* (wild-type; *swc5⁺*) strains. For genes linked to the *swc5* locus (hotspot targets; 871 genes), expression in Δ*swc5* was significantly more correlated with *swc5-fs* segregants (red) than with *swc5⁺* segregants (blue) (*$P = 1.3 \times 10^{-11}$, Wilcoxon's signed-rank test). For genes not linked to the *swc5* locus (non-targets; 5,593 genes), there was no difference in correlation. Thus, the *swc5* deletion mimics the expression signature of the Y0036 allele at this locus.

D  Average Pht1 (H2A.Z) occupancy at the 5′-ends of all coding genes in the parental strains of the recombinant yeast library. Genes were aligned at the transcription start site. Pht1 peaks at the +1 nucleosome after the transcription start site. There is significantly less Pht1 at the +1 nucleosome in Y0036 than in the 968 background ($P < 10^{-15}$, one-sided Wilcoxon's signed-rank test, n = 5,016).

E  Comparative average Pht1 occupancy in 968 (*scw5⁺*) of the genes linked (orange, n = 602) and not linked (gray, n = 4,504) to the *swc5* locus. There is significantly more Pht1 at the +1 nucleosome in genes associated with the *swc5* QTL than in others ($P < 10^{-15}$, one-sided Wilcoxon's rank-sum test).

## swc5-fs is associated with an increase in antisense and a reduction in sense RNAs

It has been shown that H2A.Z was involved in the repression of antisense transcription (Zofall *et al*, 2009; Ni *et al*, 2010). Since *swc5-fs* was leading to a reduction of H2A.Z occupancy, an increase in the antisense RNA levels was expected in *swc5-fs* strains. Therefore, we explored the direction of the genetic regulation of *swc5-fs* on its targets for sense and antisense transcription. For coding genes, hotspot 8 caused a striking increase in antisense RNA levels and a reduction in sense RNA levels. For non-coding genes, on the other hand, the reverse trend was observed (Fig 6A), consistent with the fact that many ncRNAs in fission yeast correspond to antisense transcripts of coding genes (Wilhelm *et al*, 2008; Ni *et al*, 2010; Bitton *et al*, 2011; Rhind *et al*, 2011). Similar observations have been made in strains deleted for H2A.Z, Swr1, or RNAi components (Zofall *et al*, 2009), thus suggesting that the same, or similar, molecular mechanisms are involved in *swc5-fs*. Antisense transcription can negatively affect the expression of the corresponding sense transcripts through several mechanisms, including transcriptional collision, RNA interference, and modification of mRNA stability (Faghihi & Wahlestedt, 2009; Ni *et al*, 2010; Bitton *et al*, 2011;

Chen *et al*, 2012b). Accordingly, we observed that a significant fraction of *swc5* eQTL gene targets whose sense expression was down-regulated in *swc5-fs* were also *swc5* aseQTL targets ($P < 10^{-15}$, one-sided Fisher's exact test) and *swc5* eQTL targets whose sense expression was up-regulated were unlikely targets of corresponding aseQTLs ($P = 0.006$, Fig 6B). Hence, sense and antisense transcription seem to be directly coupled for at least a subset of genes, with increased antisense transcription coinciding with decreased sense transcription, as previously observed (Faghihi & Wahlestedt, 2009; Xu *et al*, 2011; Chen *et al*, 2012a; Pelechano & Steinmetz, 2013). We confirmed these results by semi-quantitative real-time PCR (RT–qPCR) with strand-specific reverse transcription in four different genes which sense and antisense levels were linked to the hotspot 8 (Fig 7, Supplementary Fig S16). For those genes, we showed an increase in antisense-to-sense ratio in a pool of segregants carrying wild-type *swc5* compared to a pool of segregant carrying *swc5-fs*. Moreover, we also observed these differences among the parental strains in all cases but one (Fig 7A, Supplementary Fig S16B). This constitutes a clear example of transgressive segregation and highlights that, at least for this gene (*cdb4*), other genetic factors influence antisense transcription, as detected in the aseQTL analysis.

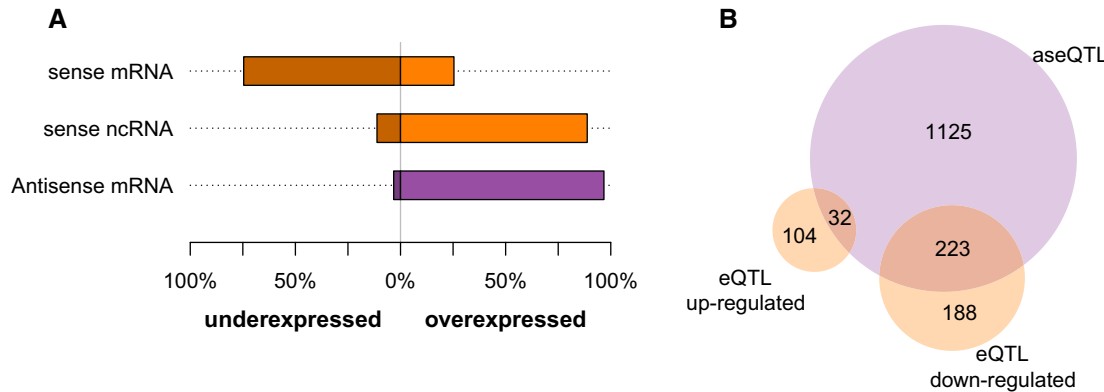

**Figure 6.  Influence of *swc5* locus on sense and antisense expression.**

A   Directionality of the effect of *swc5-fs* on expression for eQTL (orange) and aseQTL (purple). Coding (mRNAs) and non-coding (ncRNAs) transcripts are distinguished. Fractions of up- and down-regulated traits (relative to strains carrying the *swc5*⁺ allele) are shown.

B   Overlap between up- and down-regulated eQTL targets and aseQTL targets of the *swc5* locus. The overlap between down-regulated eQTL and aseQTL targets is highly significant (*P* = 2.2e-16, one-sided Fisher's exact test), whereas there is a significant depletion of down-regulated eQTL targets among the aseQTL targets (*P* = 0. 006302, one-sided Fisher's exact test).

## Possible molecular mechanisms underlying antisense transcription in *swc5-fs* strains

Possible molecular mechanisms leading to antisense transcription include read-through transcription (improper transcription termination), bi-directional transcription, overlapping transcripts, and autonomous antisense transcription (Ni *et al*, 2010; Bitton *et al*, 2011; Chen *et al*, 2012b). Read-through transcription occurs when two genes are oriented in a convergent (tail-to-tail) manner. H2A.Z has been shown to specifically play a role in the suppression of antisense transcripts originating from convergent genes in fission yeast (Zofall *et al*, 2009; Ni *et al*, 2010; Anver *et al*, 2014). In this configuration, the transcript of one of the gene (overlapping or read-through; note that in *S. pombe*, this distinction is questionable, Supplementary Information) becomes the antisense transcript of the neighboring gene encoded on the opposite strand (Zofall *et al*, 2009; Ni *et al*, 2010). In particular, deletion of H2A.Z (Δ*pht1*) causes widespread increase in antisense levels originated from convergent genes (Zofall *et al*, 2009; Kumar *et al*, 2013). Since we showed that *swc5-fs* was leading to reduced H2A.Z occupancy, we hypothesized that molecular mechanisms leading to increased antisense levels in targets of *swc5* also implicated a defect in the suppression of antisense RNAs emanating from read-through transcription. We performed additional analysis and experiments to assess this hypothesis.

First, we noticed that 72 % of the antisense traits linked to the *swc5* hotspot were antisense of convergent genes. A more precise analysis confirmed an important and significant enrichment of convergent genes among the *swc5* antisense targets (*P* < 10⁻⁶, Fig 8, Supplementary Fig S17, Supplementary Information). This result is consistent with the effects of the genetic variation at hotspot 8 going through H2A.Z.

Second, we performed RT–qPCR in the four previously studied genes in which sense and antisense were linked to hotspot 8 (see above) in strains deleted for *swc5*, H2A.Z (Δ*pht1*), and in a Δ*pht1* Δ*clr4* double mutant. The H3K9 methyltransferase Clr4 is partially

redundant with H2A.Z with regard to antisense suppression (Zofall *et al*, 2009; Zhang *et al*, 2011), and their co-mutation has been shown to synergistically decrease antisense suppression (Zofall *et al*, 2009). Results (Fig 7, Supplementary Fig S16B) show that the antisense-to-sense ratio was increased in these mutants, with the exception of Δ*swc5* for two of the four examples (for those two genes, we attribute the lack of consistency in the *swc5*-knockout to other variants in linkage disequilibrium with *swc5*, or effects of auxotrophic markers in the knockout strain, Supplementary Information). Moreover, compared to Δ*pht1* (H2A.Z): (i) effects were weaker in Δ*swc5*, as expected since *swc5* leads to a reduction of H2A.Z occupancy, (ii) effects were stronger in a Δ*pht1* Δ*clr4*, as expected because of their synergistic effect on antisense RNA suppression (Zofall *et al*, 2009). These results are also consistent with our hypothesis (Supplementary Information).

Since read-through transcripts were observed in absence of H2A.Z (Zofall *et al*, 2009; Ni *et al*, 2010), we tried to characterize their presence. We performed 3′-rapid amplification of cDNA ends (RACE) experiments in the convergent gene pair *its3-tpp1* to detect potential read-through originating from *tpp1* promoter (Supplementary Fig S18). Although read-through transcripts were detected in all tested strains (RS3, RS4 and RS5 in Supplementary Fig S18), signals were stronger in Y0036, Δ*swc5*, Δ*pht1*, and other mutants known to affect read-through transcription compared to signals in 968 and other control strains (Supplementary Fig S18B and C). This result is also consistent with our hypothesis.

Although not formally conclusive, these results taken together strongly support the hypothesis that *swc5-fs* is the major molecular event at hotspot 8 and that it influences the levels of multiple RNAs via a compromised H2A.Z deposition. This reduction of H2A.Z would lead an increase in antisense levels because of the non-suppression of antisense transcript at convergent genes pairs (overlapping transcripts or read-through transcripts), which in turn negatively affects the levels of the sense counterpart. This sequence of molecular events could directly explain 68 % of the antisense expression effects (proportion of aseQTL targets of *swc5* whose

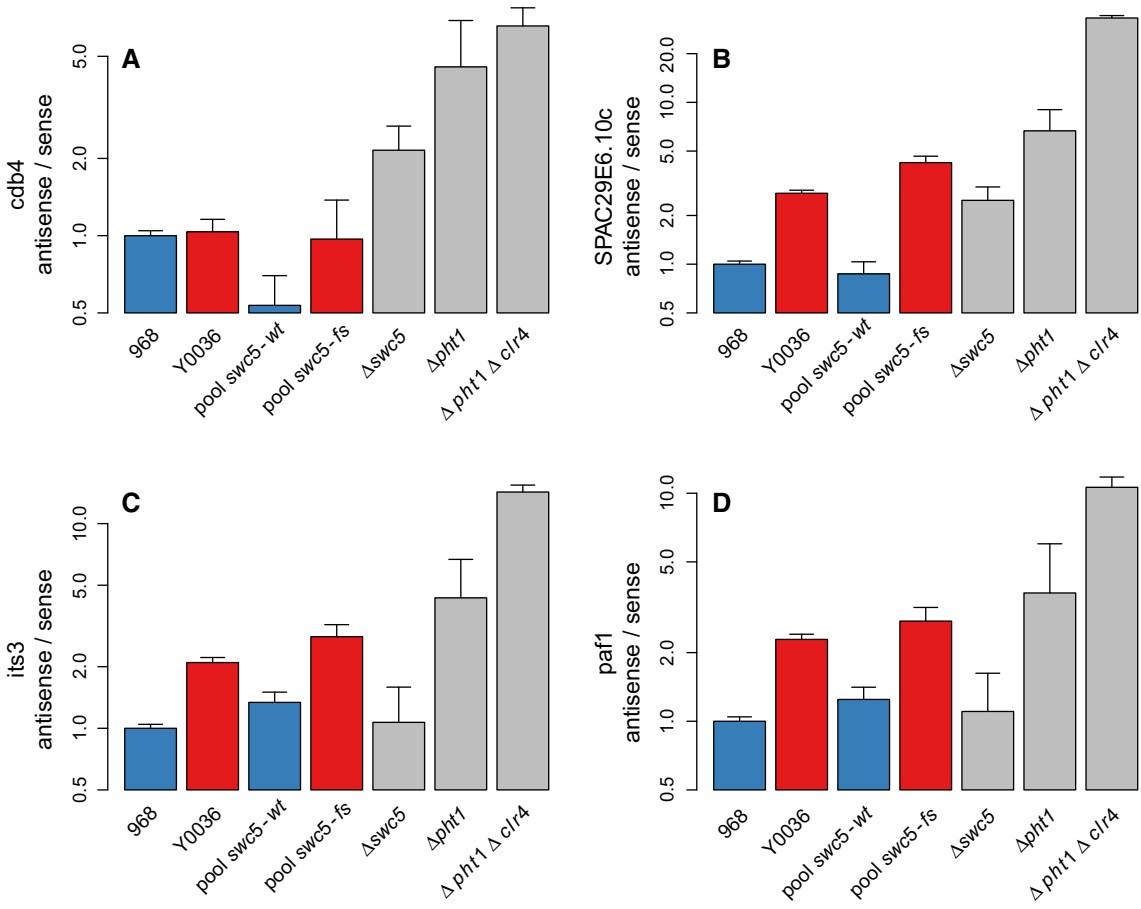

**Figure 7. Strand-specific semi-quantitative real-time PCR quantification of sense and antisense levels of selected targets of *swc5* locus.**

A–D  *y*-axis shows the ratio of antisense-to-sense levels relative to ratio in 968 strain (log scale). Error bars indicate the standard error to the mean among the three biological replicates. Analysis was performed for four genes: *cdb4* (A), *SPAC29E6.10c* (B), *its3* (C), *paf1* (D) in seven different strains: the two parental strains (968, Y0036), two pools of strains corresponding to all the segregants separated according to their genotype at the *swc5* locus, and strains deleted for *swc5*, *pht1*, and both *pht1* and *clr4*. No difference was observed between the parents in *cdb4*, which indicates that other genetic events are implicated, as expected since *cdb4* is also associated with another aseQTL (Supplementary Information). For *paf1* and *its3*, effects observed in *swc5-fs* strains (pool and Y0036) were not observed in Δ*swc5*. This may be due to other genetic effects in *cis* of *swc5*, or to effects of auxotrophic and mating makers which differ between 968 and Δ*swc5* (Supplementary Information).

expression is increased and that are part of convergent gene pair) and 31% of the sense expression effects (proportion of eQTL targets of *swc5* whose expression is decreased and that are part of convergent gene pair) of the major eQTL hotspot on Chromosome III.

## Discussion

We have generated a genetically and phenotypically diverse fission yeast strain library of segregants from a cross between 968 (standard laboratory strain) and Y0036 (independent natural isolate). The widely used RMxBY library in budding yeast (Brem *et al*, 2002; Yvert *et al*, 2003; Bloom *et al*, 2013) has proven to be an excellent tool for studying the relationship between genotype variation and complex traits (Ehrenreich *et al*, 2009; Liti & Louis, 2012). The fission yeast library will be an important complement, because compared to budding yeast, many characteristics of fission yeast are

more similar to metazoan cells. The importance of non-coding RNA and fission yeast's chromatin structure are just two examples (Käufer & Potashkin, 2000; Schwartz *et al*, 2008; Creamer & Partridge, 2011).

Conducting a high-power RNA-seq eQTL study in this strain library enabled us to: (i) detect a much larger fraction of *trans*-eQTLs than previous studies, (ii) systematically explore eQTLs affecting coding, non-coding, and antisense transcripts, (iii) reveal that ncRNA expression is at least as strongly affected by natural genetic variation as mRNA expression, (iv) detect eight hotspots that are linked with the expression variation of thousands of transcripts, both coding and non-coding, (v) uncover a genetic variation of *swc5* that modifies the levels of ~800 sense and ~1,400 antisense transcripts, and (vi) accumulate evidence pointing at elevated read-through transcription via compromised deposition of the histone variant H2A.Z as the underlying molecular mechanism.

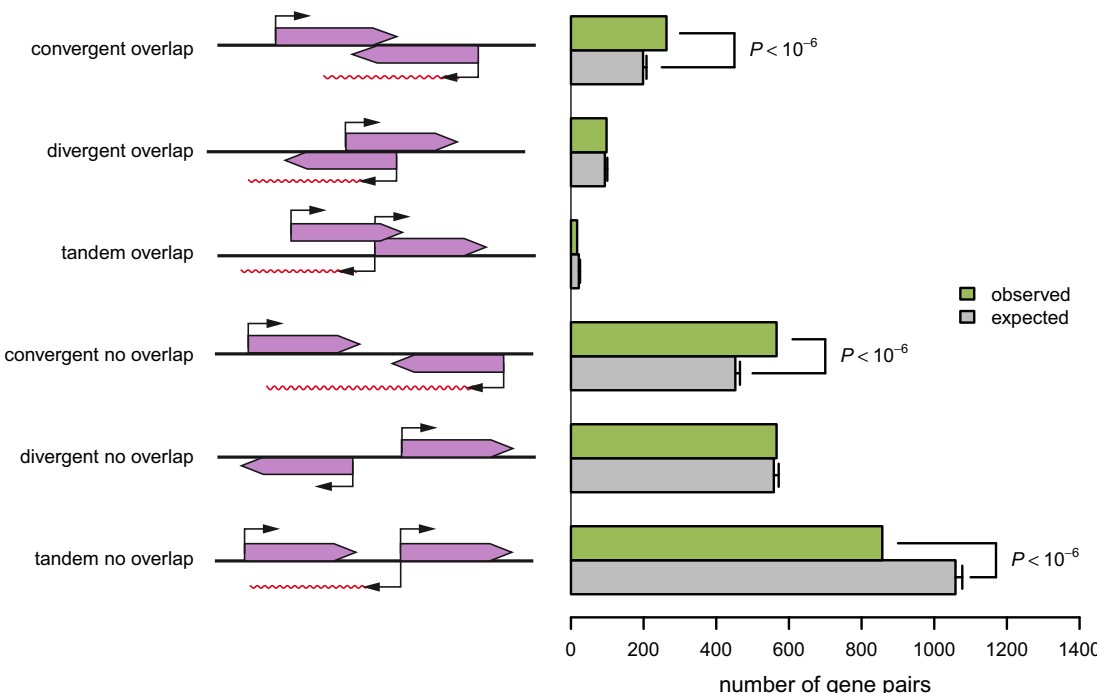

**Figure 8. Organization of coding genes whose antisense expression is linked to the *swc5* locus.**

The organization of gene pairs in the genome is either convergent, divergent, or in tandem and overlapping or not. However, the distinction between overlapping and non-overlapping genes is questionable in *Schizosaccharomyces pombe*, because of imprecise definition of the untranslated regions (UTRs). Scheme on the left shows geometry of gene pair orientations and the potential antisense RNA (represented in red wavy lines). The number of *swc5* aseQTL targets falling into each category is shown (green; observed). We used an empirical distribution based on 1,000,000 random assignments of antisense *swc5* targets (gray; expected), to test for enrichment or depletion of each category (*$P < 10^{-6}$). The depletion of tandem gene pairs is due to the non-random organization of the *S. pombe* genome (Supplementary Information, Supplementary Fig S17). Error bars indicate standard deviation over 1,000,000 permutations.

One of the most surprising results of this study was the small fraction of *cis*-eQTLs that were detected (only 10.4% of the eQTL were *cis*-eQTL). Even after excluding hotspot 8 and the Chromosome I inversion, we still observed substantially more *trans*- than *cis*-eQTL (14.0% *cis*-eQTL). This observation could partly be explained by considering strain-specific genotypes, which reduces the potential for false-positive *cis*-eQTL calls. However, although our analysis confirmed that accounting for individual genomes improves transcript quantification, the magnitude of effects in this study was far too small to fully explain the drastically reduced fraction of *cis*-eQTLs. A second explanation could be a too restrictive definition of *cis*-QTL. Our definition of a 'cis-region' potentially affecting a gene is based on the correlation of flanking markers (i.e. the extent of linkage disequilibrium; see Materials and Methods). However, lowering the threshold of correlation even further (to 0.7 or 0.6) did not change our conclusion (12.5 and 14.9% of *cis*-eQTL, respectively). A third explanation is the established phenomenon that increasing statistical power of an eQTL study increases the number of detected *trans*-eQTLs, whereas the number of detected *cis*-eQTLs starts to plateau (Schadt *et al*, 2003; Doss *et al*, 2005; Rockman & Kruglyak, 2006; Gerrits *et al*, 2009; Ackermann *et al*, 2013). Thus, under the assumption that the fraction of *trans*-affected genes in our cross is not substantially higher than in other comparable crosses, this observation suggests a high statistical power of our study. This increase in statistical power could be due to a combination of several factors. First, the genetic similarity of the strains reduces the complexity of the traits, thus simplifying the investigation of its genetic basis, notably by reducing the problem of multiple-hypothesis testing (McClurg *et al*, 2007). Note that increasing genetic similarity reduces the total number of *cis*-eQTL and also statistically simplifies the detection of eQTL. Second, the use of deep sequencing may have reduced the data noise, thus facilitating the identification of smaller genetic effects. Indeed, previous studies suggest that RNA-seq can be more precise than DNA microarrays, especially for highly or lowly expressed genes, due to its larger dynamic range (Mortazavi *et al*, 2008; Degner *et al*, 2009; Wang *et al*, 2009; Marguerat & Bähler, 2010; Malone & Oliver, 2011) (but see Willenbrock *et al*, 2009; Malone & Oliver, 2011; McIntyre *et al*, 2011). This notion is supported by our data, because we saw no bias with regard to gene expression level for the detection of eQTLs. Consistent with other recent work performed using mouse eQTL data (Ackermann *et al*, 2013), our findings suggest that the fraction of *cis*-eQTLs may have been overestimated in previous studies, due partially to lacking statistical power for the detection of *trans*-eQTLs. We wish to note, however, that while the fraction of *cis*-eQTL depends partially on the statistical power of the study, the absolute number of *cis*-eQTL that could be detected will largely by determined by the genetic similarity (or difference) of the strains being crossed.

The RNA-seq-based genotyping method reduces the costs and workload at the bench, since no additional genotyping experiments are required. The method has great potential for the application to

other recombinant strains, where parental strains have been fully sequenced (Keane *et al*, 2011). This approach requires relatively deep sequencing to be efficient, but its accuracy is robust against changing the sequencing depth (Supplementary Fig S19, Supplementary Information). The sensitivity of RNA-seq-based genotyping was higher than most other strategies used for genotyping recombinant populations (Supplementary Information).

Natural genetic variation is known to bias expression quantification based on both microarrays and RNA-seq, with known impacts on eQTL studies (Alberts *et al*, 2005, 2007; Degner *et al*, 2009; Verdugo *et al*, 2010). The fact that RNA-seq also indirectly sequences the actual genome of an individual enabled us to pursue a two-step strategy, where we first used the RNA-seq data for determining an individual's genotype to subsequently map against this individualized genome to quantify expression. Previous studies have used similar strategies for other applications requiring sensitive transcript quantification (Degner *et al*, 2009; Turro *et al*, 2011; Reddy *et al*, 2012; Satya *et al*, 2012; Franzén *et al*, 2013; Pandey *et al*, 2013; Stevenson *et al*, 2013), and here, we evaluated the relevance of this phenomenon for eQTL mapping. Our analysis, using real and simulated data, revealed significant improvements of small magnitude when accounting for the strain-specific genome, with, however, minimal impact on eQTL mapping results. These differences are likely to become more prominent in RNA-seq studies involving genetically more diverse individuals such as human populations and especially cancer genomics (Supplementary Fig S8D and E). We therefore also advocate the development of read-mappers that incorporate sequence variations into their model, which would solve the issue of our strategy with respect to homologous regions (Supplementary Information; Degner *et al*, 2009; Wu & Nacu, 2010; Reddy *et al*, 2012).

Another important advantage of RNA-seq is its ability to quantify all transcribed products, without relying on previously determined annotation (Wilhelm *et al*, 2008; Wang *et al*, 2009; Costa *et al*, 2010; Marguerat & Bähler, 2010). In the context of this study, we could show that ncRNA levels are even more affected by eQTLs than mRNA levels. Previous eQTL work only considered the *cis* regulation of a subset of the large intergenic ncRNAs (Kumar *et al*, 2013). Here, RNAs were enriched for polyadenylated species, which make up the great majority of ncRNAs in fission yeast (Marguerat *et al*, 2012). By using protocols especially developed for the purification of ncRNAs, one might detect even more ncRNA-eQTLs.

It has been shown that erroneous antisense expression quantification could arise from reverse transcription leading to positive correlation between sense and antisense levels (Perocchi *et al*, 2007). Our results show not such correlation. In the contrary, we observe an anticorrelation between the expression level of sense transcripts and the expression levels of their antisense counterparts ($\rho = -0.34$, Spearman's rank correlation coefficient, Supplementary Fig S20), as one could expect considering the repressive *cis* effect of antisense transcripts (Pelechano *et al*, 2013).

The use of strand-specific RNA-seq provided remarkable and unforeseen advantages for analyzing the *swc5* locus. Without any additional experiments, we could investigate antisense transcription and conduct an 'aseQTL study'. Importantly, even though the majority of aseQTLs were associated with the *swc5* locus, many additional aseQTL regions could be detected. Future research may address the role and mechanisms of those loci.

Our results showed that *swc5-fs* reduced the replacement of histone H2A by the H2A.Z variant at the 5′ end of coding genes. In budding yeast, Swc5 is implicated in the transfer of H2A.Z, but not in the binding of H2A.Z to the nucleosome (Wu *et al*, 2005). Our H2A.Z ChIP-seq results suggest a similar role in fission yeast: The positioning of H2A.Z was unchanged in *swc5-fs* strains, but the occupancy, that is, the efficiency of the deposition, was reduced (Fig 5D). In accordance with previous studies of the function of H2A.Z (Zofall *et al*, 2009; Ni *et al*, 2010; Zhang *et al*, 2011), our data hint that the reduced H2A.Z occupancy at the 5′ end of coding genes increased antisense transcription either through read-through transcription or through overlapping convergent gene pairs. H2A.Z is known to cooperate with RNAi and heterochromatin factors to suppress antisense read-through, presumably via exosome-dependent degradation (Zofall *et al*, 2009; Ni *et al*, 2010) (schematized in Supplementary Fig S21). The anticorrelation between sense and antisense transcription that we observed is in agreement with the hypothesis that sense and antisense transcription establish a negative feedback cycle (reviewed in Pelechano & Steinmetz, 2013). However, the effect sizes that we observed differed between sense and antisense transcription: While *swc5-fs* caused a dramatic increase in antisense expression, the associated reduction in sense expression was weaker. Indeed, linkages between sense traits and the *swc5* locus were only found for ~19% of the *swc5* aseQTL targets, showing that there is no simple and general antagonistic relationship between sense and antisense expression (Xu *et al*, 2011; Chen *et al*, 2012a).

According to our high-density growth data, the *swc5-fs* allele substantially compromised cellular fitness, resulting in less efficient growth. It is likely that this growth defect indirectly altered the expression of some genes that were not directly affected by the H2A.Z deposition phenotype. We found 153 coding genes linked to *swc5* that were up-regulated in *swc5-fs* strains and thus likely not affected by antisense transcription. The directionality of these linkages suggests that they were indirectly affected by *swc5-fs*; indeed, there was a depletion of *swc5* aseQTLs among those genes (Fig 6B). Accordingly, these genes were also highly enriched for genes involved in stress response (Supplementary Table S5), which is typically regulated as a function of growth (López-Maury *et al*, 2008).

In addition to the library of segregants, the datasets generated in this project provide a rich resource for future studies. Many aspects of the data have not been analyzed so far, such as the impact of sequence variation on alternative transcription start or polyadenylation sites, or exon usage. As opposed to budding yeast, a substantial number of fission yeast genes are spliced. The depth and precision of this data makes it an excellent resource for addressing these questions.

# Materials and Methods

### QTL strain library construction

To cross different strains and select for segregants derived from a mating between the two parental strains, we introduced dominant markers at the *ade6* locus in Chromosome III. We deleted differentially the *ade6* locus in the two parental strains by inserting

nourseothricin (NAT) or hygromycin B resistance markers at this locus (Gregan *et al*, 2006). Subsequently, we plated the mating mixture onto double selection plates that contained both NAT and hygromycin B. This strategy allowed us to select for diploid hybrids that arose from a cross between the two parental strains. F1 haploid recombinants were obtained straight from the selected diploid hybrids by tetrad analysis. F2 haploid recombinants were obtained by performing tetrad analysis of F2 diploid hybrids obtained from a mass mating among F1 haploid segregants. Double selection of dominant markers was performed to select F2 diploid hybrids, thus avoiding the high proportion of identical sibling matings observed in a mass mating involving an $h^{90}$ homothallic strain.

## High-throughput profiling of cellular growth

The growth of the parental and segregant strains was profiled in normal conditions using the BioLector micro-fermentation system (mp2p-labs GmbH, Baesweiler, Germany). Based on the light scattering (LS) technology, the BioLector system records biomass values at 620 nm (Kensy *et al*, 2009). Cells of each strain were grown for 36 h in pre-cultures, which were prepared in 10-ml falcon tubes containing YES broth (Formedium, PCM0305). The pre-cultures were used to inoculate microtiter plates with 48 'flower'-shaped wells (m2p-labs GmbH, MTP-48-B) filled with YES broth. Compared to round or square wells, 'flower'-shaped wells provide improved mixing and oxygen transfer within the culture (Funke *et al*, 2009). The cell density in each well of the plate was adjusted to an optical density (OD$_{595}$) of 0.2, with the final culture volume being 1.5 ml. Gas-permeable adhesive seals (Thermo Fisher Scientific, AB-0718 and m2p-labs GmbH, F-R48M-25) were used to cover the wells of the plate. Duplicate cultures were prepared for each strain tested.

Micro-fermentations were performed with the following settings: Temperature was set at 32°C, humidity at 99%, shaking at 1,000 rpm, and LS measurements were retrieved every 3 min. A linear regression approach between the LS and OD$_{595}$ data measured in the beginning and at the end of each experiment was applied to convert the LS data of a BioLector experiment to OD$_{595}$ values. The raw growth data are provided in Supplementary Dataset S1.

## Estimation of growth parameters

The collected BioLector data constituted a useful resource for the detailed phenotypic characterization of the recombinant library in normal growth conditions. Diverse growth parameters were extracted from the highly resolved growth curves. Firstly, we calculated the growth efficiency of the segregants, which defined by the gain of biomass given the substrate provided in the medium. The efficiency of growth was simply the change in optical density (OD$_{595}$) value of the culture from the time of inoculation to the end of the experiment (i.e. stationary phase) (Supplementary Fig S1B). Secondly, the area under each growth curve, which is another indicator of the growth capacity of a given strain, was calculated after subtraction of a baseline (Supplementary Fig S1C). Moreover, calculation of the maximum slope of the log-transformed OD-calibrated values within the exponential phase provided an estimation of the maximum specific growth rate ($\mu_{max}$). The $\mu_{max}$ of a culture was also estimated from the log-scale OD-calibrated values via the

first derivatives of a smoothing spline fit. Furthermore, we calculated the lag time (lagT), which corresponds to the time at which the tangent to the maximal growth rate intersects the time axis (Supplementary Fig S1D). Finally, we estimated the doubling (or generation) time of a culture ($T_d$), which provides a direct link between growth and cell cycle and is inversely proportional to the growth rate ($T_d = \ln(2)/\mu_{max}$). Growths parameters are provided in Supplementary Dataset S2.

## Cell survival at the stationary phase of growth

Aliquots of biomass samples collected from stationary phase cultures were diluted to ~$10^6$ cells, and a 1/10 aliquot was stained with 0.05 g/l propidium iodide (Sigma, P4170). Stained cells were incubated for 5 min at room temperature and analyzed for cell survival using a Cell Lab Quanta SC MPL flow cytometer (Beckman Coulter, Fullerton, CA). The jumper was set to 'small', and the cell survival protocol of the Cell Lab Quanta SC software was applied with the following gain and voltage settings: EV = 4.80, SS = 5.00, FL1 = 4.22, and FL2 = 4.13. Flow cytometry data were analyzed using the FlowJo v.7.1 (Tree Star, Inc.) software package. Survival data are provided in Supplementary Dataset S3.

## Parental strain genome sequencing

DNA was extracted with Qiagen genomic DNA kits. Paired-end Illumina libraries were constructed according to the manufacturer's instructions, with 300 to 400 nt fragmented DNA. Sequences were produced using a Illumina Genome Analyzer II, to at least 240× coverage. Sequence data are archived in the European Nucleotide Archive (http://www.ebi.ac.uk/ena/) under the accession numbers ERX007392 and ERX007395 for the strains 968 and Y0036, respectively.

## RNA-seq library preparation and sequencing

Strand-specific RNA-seq libraries were created for SOLiD sequencing from poly(dT)-enriched RNA using the SOLiD™ Total RNA-Seq kit (Applied Biosystems, LifeTechnologies). Briefly, 50-ml cultures grown at 32°C were harvested at 0.4–0.5 OD$_{595}$ by filtration and snap-frozen in liquid N2. Total RNA was isolated by hot phenol extraction (Lyne *et al*, 2003), and RNA quality was assessed on a Bioanalyzer instrument (Agilent). For poly(dT)-enrichment, three rounds of poly(dT) Sera-Mag magnetic bead purifications were carried out using 50 µg of total RNA starting material and verified on a Bioanalyzer. 500 ng poly(A)-enriched RNA was then fragmented to an average size of ~200 nt. SOLiD adaptors were hybridized and ligated, RT–PCR performed, and the cDNA purified by size selection on 6% TBE-urea gels. The 3′ barcodes (1–48) were incorporated at the library amplification step using 16 PCR cycles and purified using SPRI-beads (Agencourt, Beckman Coulter). Library size distributions and concentrations were determined on a Bioanalyzer, and samples were pooled and put through emulsion PCR steps. RNA-seq libraries were sequenced on an ABI SOLiD V4.0 System (50 bases long reads; Applied Biosystems, LifeTechnologies).

The RNA-seq data are available at ArrayExpress (http://www.ebi.ac.uk/arrayexpress/, identifier E-MTAB-2640).

                    

## ChIP-seq library preparation and sequencing

To investigate H2A.Z (Pht1) occupancy in 968 compared with Y0036 cells, endogenous *pht1* was c-terminally tagged in 968 and Y0036 strains with 13c-myc:kanMX6 (Bähler *et al*, 1998). We named the resulting strains JB1221 and JB1220 (*968-pht1-13c-myc* and *Y0036-pht1-13c-myc*, respectively) and verified the successful tagging by Western analysis, Supplementary Fig S22A). The *swc5Δ-pht1-13c-myc*-tagged strain was created by crossing JB1221 with a prototroph *swc5Δ* strain. We used the *h+ swc5Δ* strain from the *S. pombe* Bioneer deletion library (Kim *et al*, 2010), crossed out all the auxotrophic markers associated with that background using the *h⁻* 972 wild-type strain, and then crossed the prototroph *swc5Δ* with the previously generated *968-13c-myc* strain. Two *swc5Δ-pht1-13c-myc*-tagged clones were verified by PCR and Western analysis (Supplementary Fig S22B and C), named JB1448 and JB1449, and used for further analysis.

Western verification of C-terminal *13cmyc-tagged pht1* strains (Supplementary Fig S22A) was performed using the Novex NuPAGE system (LifeTechnologies). Cell lysates, 15 μg, were separated on 4–12% Bis–Tris Gel, transferred to nitrocellulose membrane, and incubated with anti-c-myc antibody (Ab32, Abcam).

To verify *swc5*-knockout, checking primers were used in colony PCR (Supplementary Fig S22C). For *swc5Δ* 5′, junction checking primers CPN1 (within KanMX) 5-CGTCTGTGAGGGGAGCGTTT-3 and Cp5 5-TAACAAATCCCCCACAAGTCTTATT-3 (upstream of *swc5* gene) were used. For 3′, junction checking primers CPC1 5-TGATTTTGATGACGAGCGTAAT-3 (within KanMX) and Cp3 5-ACAACAAGCATTGCAACATCACAAT-3 (downstream of *swc5* gene) were used.

ChIP-seq libraries were created for Illumina MiSeq sequencing from Chromatin immunoprecipitations (ChIPs) using antibodies specific for either c-myc or histone H3 (ab32 and ab1791, respectively; Abcam) in biological triplicates. Briefly, 500-ml cultures grown at 32°C were formaldehyde-fixed for 30 min, harvested by centrifugation, and snap-frozen. Chromatin extract (CE) was prepared with a Fastprep machine (MP Biomedicals) at 6 × 20 s, 5.5 m/s using an equal volume of 0.5 ml acid-washed glass beads (BioSpec products) to break cells. Lysates were sonicated using a Bioruptor (Diagenode) at 6 × 5′, 30 s on, 30 s off, in ice-cold conditions, to create CE with an average DNA size of approximately 200 bp. Immunoprecipitations (ChIPs) were set up using 5 mg CE on Dynabeads Protein A with appropriate antibody (Invitrogen, LifeTechnologies). Input (CE) and ChIPs were de-crosslinked overnight, treated with DNase-free RNase (Roche) and proteinase K (Invitrogen, LifeTechnologies), and the DNA-purified using PureLink PCR micro kits (Invitrogen, LifeTechnologies).

Triplicate ChIP-seq libraries from the immunoprecipitated samples and corresponding input material were prepared using the NEBNext ChIP-Seq Library Prep Master Mix set for Illumina (New England Biolabs) combined with Illumina Barcodes 1–12. Briefly, and according to manufacturer's instructions, DNA was end-repaired, end adaptors ligated, and the libraries amplified with 15 PCR cycles in the presence of Illumina barcodes. Libraries were purified using SPRI-beads (Agencourt, Beckman Coulter), and library size distributions and concentrations were determined by Bioanalyzer (Supplementary Fig S22D) and Qubit

(Invitrogen, LifeTechnologies). Pooled ChIP-seq libraries were loaded at 10 pmol onto a MiSeq instrument using reagent kit v2, 50–55 cycles (Illumina). The ChIP-seq data are available at ArrayExpress (http://www.ebi.ac.uk/ arrayexpress/ identifier E-MTAB-2650). qPCRs were performed in triplicate, as a QC to verify enrichment in ChIP samples over input material. Briefly, four regions over the *S. pombe adh1* locus, including the transcription start site, were examined using the fast SYBR Green Master Mix system (Applied Biosystems). Relative quantities were derived from $C_t$ values obtained from the SDS software (Applied Biosystems) using 10-fold serial dilutions of input material as standards. Once enrichment was verified, ChIP-seq libraries were constructed. Supplementary Fig S22E shows qPCR data over the transcription start site of *adh1*.

### Strand-specific RT–qPCR

#### Design

RT–qPCRs were performed for four pairs of genes arranged convergently. The gene pairs were *cdb4* (SPAC23H4.09)/*thi4* (SPAC23H4.10c); SPAC29E6.10c/SPAC29E6.09; *its3* (SPAC19G12.14)/*tpp1* (SPAC19G12.15c); and *paf1* (SPAC664.03)/*rps1602* (SPAC664.04c); the first indicated gene of each pair was the *swc5* target gene. For each pair of convergent genes, two different reverse-transcription (RT) reactions (plus two control reactions) and four qPCRs were performed (the procedure was repeated in triplicate). The cDNAs of all the sense transcripts were generated in one of the RT reaction, while the antisense cDNAs were generated in a second reaction (Supplementary Fig S16A).

#### RNA extraction

Total RNA was isolated by hot phenol extraction (Lyne *et al*, 2003). The residual DNA was digested with Qiagen's DNase via in column digestion using Qiagen's RNAeasy mini kit protocol. RNA quality was assessed on a Bioanalyzer instrument from Agilent. All RNAs used in this assay had a RIN (RNA Integrity Number) above 9.

#### RT reaction

cDNA was generated using primer-specific RT reactions (2 pmols per primer) according to Superscript III manual, and 3 μg of total RNA was retro-transcribed using Superscript (R) III Reverse Transcriptase (Life Technologies). To reduce the synthesis of primer-independent cDNA, which has been shown to account to up to 57% of the RT products in loci that express both sense and antisense transcripts (Haddad *et al*, 2007; Perocchi *et al*, 2007; Feng *et al*, 2012), the temperature of the RT reaction was set at 55°C (relatively high temperature) and actinomycin D was added to the reactions after the denaturating step at 70°C at a concentration of 6 μg/ml (Perocchi *et al*, 2007). For each RNA, 4 sets of RTs were carried out:

1  Sense RT.
2  Antisense RT.
3  Self-priming RT in which no primers were added to assess for the primer-independent cDNA synthesis.
4  DNase control RT in which no enzyme was added, to estimate the remaining DNA after DNase treatment.

The list of RT primers is provided in Supplementary Table S6.

*Semi-quantitative PCR*

The expression of the transcripts analyzed was performed using the fast SYBR Green Master Mix system (Applied Biosystem). $C_t$ values were obtained from the SDS software (Applied Biosystem). The list of qPCR primers is provided in Supplementary Table S6. Raw qPCR data are provided in Supplementary Dataset S16.

*Analysis*

All qPCRs were performed relative to the expression of the house-keeping gene *cdc2*. Because we were assessing expression differences induced by genetic variations that had widespread effects on transcription levels, we could not assume that the level of house-keeping genes (like *cdc2*) were stable between the strains. There-fore, we restricted our analysis to the comparison of ratios (like antisense-to-sense ratios), because this quantity was independent of potential variations in reference gene levels. Indeed, in semi-quantitative qPCR, the expression levels are measured relatively to a reference gene (here *cdc2*); for instance, the sense level of gene g ($sense_g$) is estimated via $sense_g/cdc2$ and its antisense level ($antisense_g$) via $antisense_g/cdc2$. Thus, their ratio is independent of cdc2 level:

$$\frac{antisense_g/cdc2}{sense_g/cdc2} = \frac{antisense_g}{sense_g}$$

**3′-rapid amplification of cDNA ends experiments**

3′-RACE experiments were done using SMARTer RACE kit (Clon-tech) according to the manufacturer's instructions.

**Targeted proteomics analysis of *swc5***

Samples for protein analysis were harvested from the exact same cultures as that for the RNA. Fifty milliliter of growing culture (as described above) was pelleted at 4°C, washed once in 1× PBS, and snap-frozen in liquid $N_2$. Cells were harvested by centrifugation at $2,000 \times g$ and resuspended in 100 μl of lysis buffer (100 mM ammoniumbicarbonate, 8 M urea, 0.1% Rapi-Gest™) and 100 μl of glass beads. The cells were lysed first by vortexing for $3 \times 1$ min using a multivortexer followed by soni-cation for $2 \times 30$ s. A small aliquot of the supernatant was taken to determine the protein concentration using a BCA assay (Thermo Fisher Scientific) and the protein concentration adjusted to 5 mg/ml using additional lysis buffer. 100 μg total protein was employed for targeted LC-MS analysis as recently specified (Glatter *et al*, 2012). First, disulfide bonds were reduced with 5 mM TCEP for 30 min and alkylated with 10 mM iodoaceta-mide at room temperature in the dark. Excess of iodoacetamide was quenched with 12.5 mM N-acetyl-cysteine for 10 min at 37°C. Protein samples were diluted five times with 50 mM $NH_4HCO_3$ to obtain a urea concentration of 1.6 M during digest. 2 μg of trypsin (Promega) was added to the protein sample (pro-tein to trypsin ratio = 50:1), and digestion was carried out at 37°C overnight (about 18 h). Then, the samples were acidified with 2 M HCl to a final concentration of 50 mM and incubated for 15 min at 37°C, and the cleaved detergent was removed by centrifugation at $10,000 \times g$ for 5 min. Subsequently, a mixture

containing 1 pmol of heavy labeled reference peptides (Spike-Tides_L, JPT Peptide Technologies), respectively, was spiked into each sample and subsequently C18-purified using spin columns (Harvard Apparatus) according to manufacturer's instructions. The corresponding Swc5 peptides were selected from two recent large-scale proteomic studies of *S. pombe* (Marguerat *et al*, 2012; Gunaratne *et al*, 2013). Data derived from a spectral library generated based on data-dependent LC-MS/MS analysis of the standard peptide mix were imported into Skyline version 1.4 (https://skyline.gs.washington.edu/labkey/wiki/home/software/Sky-line/page.view?name=default) to define precursor charge states and the most intense transitions. Up to 10 transitions per peptide were traced on a LTQ Velos mass spectrometer connected to an electrospray ion source (both Thermo Fisher Scientific). Peptide separation was carried out using an easy nano-LC systems (Thermo Fisher Scientific) equipped with a RP-HPLC column (75 μm × 37 cm) packed in-house with C18 resin (ReproSil-Pur C18–AQ, 3 μm resin; Dr. Maisch GmbH, Ammerbuch-Entringen, Germany) using a linear gradient from 96% solvent A (0.15% formic acid, 2% acetonitrile) and 4% solvent B (98% acetonitrile, 0.15% formic acid) to 35% solvent B over 90 min at a flow rate of 0.2 μl/min. The data acquisition mode was set to obtain only MS/MS scans in the linear ion trap of the defined precursor masses. Maximal ion time was set to 50 ms, automatic gain control target was set to 30,000 ions, and one microscan was acquired per MS/MS scan. The MS data were imported into Skyline that was used for further visualization, transition detec-tion, and calculation of transition ratios. The generated transition list used for the targeted analysis and the skyline results are provided in the Supplementary Dataset S17.

**DNA re-sequencing data processing**

Paired-end reads were aligned against the *S. pombe* reference genome using BWA (Li & Durbin, 2009) (maximum edit distance 4, maximum gap expansion 15, seed length 32). We then applied the GATK pipeline (v.1.0-6148) for quality score recalibration, indel realignment, SNP, and small indel discovery and genotyp-ing across the three parental samples simultaneously using standard hard filtering parameters with a minimum variant score of 50 (DePristo *et al*, 2011). Although the strains were haploid, the GATK pipeline assumed diploidy for genotyping. We therefore rejected polymorphisms that were genotyped as hetero-zygous. We identified 4,570 polymorphisms (Supplementary Dataset S6).

**RNA-seq-based genotyping**

*Step 1: Identifying genomic variants in the segregants*

For every sample, reads were mapped to the *S. pombe* reference genome, in which the sequences of the spikes were added, using Bowtie v.0.12.7 (Langmead *et al*, 2009) (using the following command line options: −C −n 3 −e 100 –best). Read group information was added, and sample-specific BAM files were sorted using Picard utilities (http://picard.sourceforge.net/). We then used the GATK pipeline (v.1.0-6148) to perform a local realignment of reads around the indels that were previously identified in progenitor strains. In the same step, data of samples derived from identical

segregant strains were pooled together to obtain strain-specific BAM files. Next, SNPs and indels were genotyped at the sites polymorphic in the progenitors using GATK.

### Step 2: Inferring genotype

For every polymorphic site between the progenitors, we compared the polymorphisms in the segregant and the parental strains to infer which allele was inherited. Ambiguous genotype calls (heterozygous calls—we studied haploid individuals) and calls where the GATK genotype score was below 20 were considered as unknown. This threshold was determined empirically by analyzing the genotype calls made from RNA-seq data of the parental strains—for which we knew the expected genotype.

### Step 3: Filtering for potential genotyping errors

Polymorphic sites that were not correctly called in the parental strains and with a minor allele frequency below 10% were discarded (89 polymorphisms). Another class of potentially erroneous genotype calls regards markers with genotype calls differing from the two flanking markers (47 cases in the entire strain library). Such patterns are likely to denote an erroneous genotype call when the flanking markers are close (Supplementary Information). When the distance between the flanking sites was smaller than 50 kilo base pairs (kbp), those genotype calls were considered as erroneous and corrected to match the segregation pattern of the flanking sites (34 cases).

### Step 4: Imputing missing genotypes

The genotype of less than half of the polymorphic sites could be directly deduced from the RNA-seq data. Genotypes of the remaining sites could not be determined because of the lack of read coverage resulting from low (or no) expression. However, the segregation of a polymorphism is not an independent event since the meiotic recombination generates relatively large haplotype blocks. This can be taken into account to infer missing genotypes using the data of genotyped flanking polymorphic sites. Therefore, when the informative flanking sites showed the same segregation patterns and when they were < 50 kb away, the same genotype as the flanking one was assigned to the missing value, thus assuming that no recombination event took place (Supplementary Information). When the distance was greater, or when the flanking polymorphic sites showed opposite segregation patterns, the genotype was still considered as unknown.

### Step 5: Assembling a set of markers for linkage analyses

Genotypic markers that were not called in half of the segregants after the imputation step were discarded (29 sites). Adjacent markers with the same segregation pattern across all segregants were collapsed into one unique marker, resulting in a set of 708 unique mapping genotypic markers (Supplementary Dataset S9). Thus, each marker represents a genomic interval in which all polymorphisms are in full linkage disequilibrium in the cross.

If replicates RNA-seq data of the same strain led to completely different genotypes, thus indicating probable sample inversions, the data of all replicates were removed from the analysis.

### Generation of strain-specific genome and annotation files

We used the genotype data to generate both strain-specific genome sequences and strain-specific annotations starting from: (i) the reference genome (RG) sequence, (ii) the RG annotation files, and (iii) the genomic variations information (VCF file) of the strain of interest. Annotation files had to be generated to take insertions and deletions into account. For this task, we developed a standalone Python script called GenomeGenerator, which is freely available (www.cellularnetworks.org).

### RNA-seq simulation

To explore the potential advantage of the strain-specific RNA-seq mapping, we simulated sets of 48 bp long RNA-seq reads for the two parental strains (968 and Y0036) and three segregants (R1-10, R1-13, and R1-22). First, we simulated the expression levels using the average normalized expression values that we measured in the real data for these samples (the same expression values were used for all the simulations). To generate the number of RNA-seq reads to simulate for each gene (*NR*), the normalized log expression values (expression) were transformed to a natural scale and multiplied by 10. This value was then ceiled (number of reads are integer), and all values smaller than 1 were set to 1. This led to a total of 77,110,382 reads by simulated samples.

$$NR = \begin{cases} 1 & \lceil 10 * 2^{\text{expression}} \rceil \leq 1 \\ \lceil 10 * 2^{\text{expression}} \rceil & \text{otherwise} \end{cases}$$

For each strain, we then generated a strain-specific transcriptome (the sequence of the 6,464 coding and non-coding transcripts considered in this study) that took polymorphisms into account and added a poly-A tail of 48 bases for every transcript.

Before generating the RNA-seq reads, we built an error model to simulate sequencing errors. It has been shown that sequencing errors play an important role for the accuracy of RNA-seq-based gene expression quantification when polymorphisms were considered (Degner *et al*, 2009). In order to quantify the sequencing error, we used the RNA-seq results for strain 968 (laboratory strain isogenic to the strain from which the *S. pombe* reference genome was derived). We assumed that all mismatches between aligned reads and the reference genome represent sequencing errors. Therefore, for each read position, we evaluated the mismatch frequency and used it as a sequencing error probability in the error model. We then generated the RNA-seq reads by choosing randomly segments of 48 bases from the transcript sequences and added mismatches randomly following the previously generated error model. Note that the coordinates of the RNA-seq reads were the same for all the simulated strains in order to facilitate comparisons.

### Gene expression quantification and normalization

To avoid biases due to sequence variations in gene expression quantification, RNA-seq reads were mapped onto strain-specific genomes (sequences of the spikes were also added to the genome sequence file) using Bowtie v.0.12.7 (Langmead *et al*, 2009) with the following command line options: (-C —best -m 1). Gene expression was evaluated by counting the number of reads mapped into each

annotated element (strand specifically). For coding genes, only the coding sequences were considered to avoid the problem of variable UTR length (Pelechano *et al*, 2013).

Read counts were filtered to remove genes, for which no read was mapped in more than half of the samples. The remaining zero read counts were replaced by 0.1 to avoid mathematical errors in subsequent logarithmic transformation. Data were then sample-wise scaled to the total number of mapped reads (spike excluded from the counts).

$$\mathbf{e_{i\ scaled}} = \frac{\mathbf{e_i}}{t_i} \cdot \bar{t}$$

where $\mathbf{e_i}$ is the expression vector of the sample $i$, $t_i$ is the total number of mapped reads, and $\bar{t}$ is the average.

Finally, potential batch effects were removed by mean-centering (also known as zero-mean, or one-way analysis of variance adjustment (Luo *et al*, 2010)), and data were log$_2$-transformed.

$$\mathbf{e_{g\ norn}} = \overline{e_g} \cdot \begin{pmatrix} \mathbf{e_{g\,b1}}/\bar{e}_{g\,b1} \\ \mathbf{e_{g\,b2}}/\bar{e}_{g\,b2} \\ \cdots \\ \mathbf{e_{g\,bn}}/\bar{e}_{g\,bn} \end{pmatrix}$$

where $\mathbf{e_g}$ is the expression vector of the gene $g$ over all samples and $\mathbf{e_{g\,b1}}$, the expression vector of the gene $g$ over the samples of the first batch. For the computation of the mean expression across a batch ($\bar{e}_{g\,b_i}$), the samples corresponding to the *swc5* deletion strain ($\Delta swc5$) were not taken into account because they were likely to be different from the recombinant lines (different genetic background). This did not alter the analyses that were carried out with these particular samples since only correlations were studied.

## Antisense expression quantification and normalization

Antisense expression data were quantified and normalized as the sense expression data, but raw expression measurements were obtained by counting the reads mapped on the opposite strand of each annotated elements. Only the antisense expressions of coding gene were considered in all subsequent analyses. Moreover, zero values were not replaced by 0.1 but considered as missing values, to account for the overall lower read coverage of antisense traits.

## Differential gene expression analysis

Differential gene expression analysis has been carried out to compare the expression of the 6,464 sense traits and of the 4,133 antisense traits in the *swc5*-deletion strain (measured in triplicate) and the wild-type parental strain 968 (measured in seven replicates). We used DeSeq2 to carry out this analysis. DeSeq models read counts using negative binomial distributions and uses generalized linear models to test for differential expression.

## eQTL, aseQTL, and growth QTL mapping

A previously developed QTL detection method based on Random Forest (RF) (Michaelson *et al*, 2010) was adapted to handle missing genotype values. The RF-based mapping uses genetic markers as

predictors. The original RF cannot handle missing values in the predictor matrix. A naïve solution would be to define missing values as a third category (i.e. possible 'genotypes' would be the two parental alleles 968 and Y0036, and 'missing'). However, we observed that treating missing values like a genotype introduces a bias in some special cases even though the total number of missing values was very small. We thus developed a new strategy that only uses real genotypes: In the case of missing values, one of the two alleles (either 968 or Y0036) is randomly assigned to that locus. This procedure is repeated 1,000 times, each time re-assigning a random genotype, and subsequently, the resulting models (forests) are combined. This strategy removed biases previously introduced by missing values.

We further modified our Random Forest strategy to correct for potential population sub-structure, by adopting previously proposed concepts that essentially include population structure as a covariate in the model (Patterson *et al*, 2006; Price *et al*, 2006, 2010; Novembre & Stephens, 2008). First, we estimated the relatedness between the strains (i.e. the kinship matrix) using the 'emma' package (Kang *et al*, 2008). Subsequently, we selected those eigenvectors of the kinship matrix corresponding to the top eighth eigenvalues as covariates for the QTL mapping (additional predictors for growing the Random Forests). These eight-first vectors explained more than 80% of the genotype variance.

For the eQTL and aseQTL mapping, forests of 16,000 trees (100 forests of 160 trees) were grown using the R implementation of the Random Forest algorithm (randomForest package). The strategies described above were used to handle missing genotype values and model the population structure. The *mtry* parameter, which defines the number of randomly preselected predictors at each split, was left to default (one-third of the total number of predictors). The QTL was then scored using the predictor selection frequency as previously proposed (Michaelson *et al*, 2010). To estimate the significance of the linkages, each trait was permuted 1,000 times and random forests were grown for each permutation. The correspondence between the covariates (batches and population structure models) and the permutated traits was maintained in order to properly estimate the significance of the trait-marker linkages. Note that the same permutation scheme was used for all traits in order to account for inter-trait correlation. This permutation strategy has previously been proposed in order to avoid biases in the subsequent detection of eQTL hotspots (Breitling *et al*, 2008). By pooling the results of all the studied traits, we obtained null distributions of the selection frequencies for each marker (predictor). These distributions were used to generate empirical *P*-values for the selection frequencies. We then reused the permutation results to generate *P*-values for each randomized trait and thus obtained a null distribution of *P*-values that was used to estimate false discovery rates (FDR). For growth QTLs, forests of 50,000 trees were grown (2,000 forest of 25 trees) and 10,000 permutations were used to estimate the null distributions.

When consecutive markers were linked to the same trait, the QTL was considered to span over the markers of interest. Further, regions containing multiple markers linked to the same trait were combined into a single locus if the linked markers were in high linkage disequilibrium (LD) to each other (Pearson's $r > 0.8$) and separated by not more than 10 non-linked markers. All of those markers

(including the intermediate non-linked markers) were combined into one QTL (one region). If such markers were separated by more than 10 non-linked markers (or resided on different chromosomes), they were counted as a single linkage, because it is *a priori* unknown, which of those regions contains a causal variant. We refer to these sets of markers as 'QTL groups', because they do not constitute a contiguous genomic region (i.e. not a single 'locus'). QTL genomic coordinates (Supplementary Datasets S11 and S13) were reported as spanning in between the first non-linked polymorphisms. Linkages with last (or first) markers at the end (or beginning) of a chromosome were considered as enclosing the entire telomere.

### eQTL hotspot detection

To formally identify eQTL hotspots in our dataset, the genome was divided into 50-kb bins (250 bins, bins at the end of chromosomes were bigger) and the number of eQTLs falling in each bin was counted. Nineteen bins contained more eQTLs than expected ($P < 8\text{e-}4$) if the eQTLs were randomly distributed across the genome. The expected number of eQTLs per bin was computed assuming a Poisson distribution (Brem *et al*, 2002). Consecutive bins sharing the same targets were subsequently regrouped in 8 eQTL hotspots.

### *cis* versus *trans*-QTL

eQTLs and aseQTLs were considered to act in *cis* when the marker defining the QTL that was the most correlated with the markers surrounding the target gene was correlated at more than 0.8 (Pearson's *r*). The results that we obtained regarding the proportion of *cis*-eQTLs were not importantly affected by lowering the threshold (Discussion).

### Identification of the strongest candidate regulator genes

eQTL loci usually span multiple genes. Identifying the causal sequence variation or the causal gene responsible for a QTL has proven to be difficult (Bing & Hoeschele, 2005; Kulp & Jagalur, 2006; Suthram *et al*, 2008; Verbeke *et al*, 2013). Several methods take into account the correlation between the expression of the target gene and the genes located at the eQTL (regulator genes) to address this issue (Bing & Hoeschele, 2005; Kulp & Jagalur, 2006). Here, we used a similar approach to identify potential likely regulator gene for each eQTL target. For each *trans*-eQTL (or eQTL group), we computed the squared Pearson's *r* between the target gene and all annotated genes located at the QTL. The candidate regulator showing the maximal correlation was then considered as the potential regulator of the coding or non-coding target of interest. Note that this method selects one candidate regulator for each gene–QTL linkage. Therefore, if several genes are linked to the same QTL, this method can identify several strongest regulator candidates in this QTL, provided it contains several genes.

The prediction of the putative regulator at the hotspot (Supplementary Table S3) was done based on a voting approach: For each target of the hotspot, the strongest candidate regulator gene has been determined. The most frequently selected candidate regulator was considered the regulator of the hotspot.

### Directionality of the effects

To determine the directionality of the regulation of a QTL on a trait, we simply compared the average trait level in the strains carrying the Y0036 allele at the locus of interest to the average level in the strains carrying the 968 allele. The directionality was always expressed relative to 968. Note that the directionality test implied that a significant linkage had been previously found between the locus and the trait; hence, the statistical significance had already been tested.

### Gene pair organization

Only the protein-coding genes were considered in this analysis, because many ncRNAs correspond to antisense RNAs in *S. pombe* (Wilhelm *et al*, 2008; Ni *et al*, 2010; Bitton *et al*, 2011; Xu *et al*, 2011). On each chromosome, genes were ordered according to genomic position of the transcription start site position (or, when not available, the start codon). All pairs of successive genes were then classified into six categories by analyzing the orientation of the genes (plus or minus strand) and whether they overlapped. Since we evaluated the antisense transcription considering only the region located on the opposite strand of the coding region (CDS), genes were not considered as overlapping when only their UTRs overlapped.

The significance of the gene pair enrichment was empirically determined: 1,384 antisense traits were linked to the *swc5* locus; we randomly picked the same number of traits and counted the number of picked gene pairs falling in each category. By repeating this operation 1,000,000 times, we obtained empirical null distributions of the number of gene pairs in each category. No empirical value was above the observed number of overlapping and non-overlapping convergent gene pairs, and no empirical value was below the observed number of non-overlapping tandem gene pairs.

### ChIP-seq analysis

ChIP-seq reads were aligned onto the *S. pombe* reference genome using Bowtie v.0.12.7 (Langmead *et al*, 2009) (using the following command line options: `-m 1 —best`). Pht1 peaks were called using MACS2 v2.0.10.09132012 (Zhang *et al*, 2008), using input DNA (specific for each replicate) as background control. The shift size was fixed 73 in all samples; it corresponds to the average shift size modeled by MACS2 for the studied samples.

Per base Pht1 occupancy relative to control was calculated using the MACS2 '*bdgcmp*' command using the '*divide*' method (independently for each replicate). The resulting bedgraph files were used to map all coding genes onto a meta-gene. For each gene, the upstream intergenic region and part of the coding region were analyzed within a region ranging from $-800$ to $+2800$ bp from the transcription start site with a resolution of 1 bp, similarly as described (Buchanan *et al*, 2009).

### GO enrichment

Gene Ontology (GO) enrichment analyses were performed using topGO ('weight' algorithm, maximum node size 3, Fisher test),

which takes the topology of the ontology into account (Alexa *et al*, 2006).

## Data Accessibility

Sequencing data of the parental strains are archived in the European Nucleotide Archive (http://www.ebi.ac.uk/ena/) under the accession numbers ERX007392 and ERX007395 for the strains 968 and Y0036, respectively.

All RNA-seq data generated in this study are available at Array-Express (http://www.ebi.ac.uk/arrayexpress/, identifier E-MTAB-2640).

All ChIP-seq data generated in this study are available at ArrayExpress (http://www.ebi.ac.uk/arrayexpress, identifier E-MTAB-2650).

The standalone script that generates strain-specific genome sequence and annotation (GenomeGenerator), is freely available at http://www.cellularnetworks.org.

All the other raw data, processed data, and dataset results are provided as Supplementary Datasets with this manuscript.

**Supplementary information** for this article is available online: http://msb.embopress.org

## Acknowledgements

We deeply thank Ruedi Aebersold (ETH, Zurich, Switzerland) for his advice and fruitful discussions all along this study. We thank the Liverpool Sequencing Service (Pia Koldkjaer, John Kenny, Suzanne Kay, Richard Gregory, and Christiane Hertz-Fowler at the Centre for Genomic Research, University of Liverpool) for helpful discussions and assistance with the RNA-seq protocol, libraries QC, pooling, and performing SOLiD sequencing runs. Parts of the computations were performed on the cluster of the Center for Information Services and High Performance Computing (ZIH) at TU Dresden. This study was mainly supported in AB, CTW, and JB laboratories by PhenOxiGEn, an EU FP7 research project. Research in the Bähler Lab was additionally funded by a Wellcome Trust Senior Investigator grant.

## Author contributions

FXM, SM, and JB designed the experiments. FXM generated the yeast library. SC and SM generated the sequencing libraries. SC performed the ChIP-seq analysis. MRL and FXM designed and performed the RT–qPCR experiments. MAP and CTW designed, performed, and analyzed the growth experiments. AS designed, carried out, and analyzed the proteomics experiments. MC-Z and AB designed the analyses; MC-Z and SR carried out the analysis. The manuscript was written by all authors. CTW, JB, and AB envisioned and supervised the project.

## Conflict of interest

The authors declare that they have no conflict of interest.

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
