## [Review Process File · Molecular Systems Biology]

Natural genetic variation impacts expression levels of coding and non-coding genes through antisense transcripts in fission yeast

Mathieu Clément-Ziza, Francesc Xavier Marsellach, Sandra Codlin, Manos A. Papadakis, Susanne Reinhardt, Maria Rodriguez-Lopez, Stuart Martin, Samuel Marguerat, Alexander Schmidt, Eunhye Lee, Christopher T. Workman, Jürg Bähler, Andreas Beyer

Corresponding author: Andreas Beyer, University of Cologne

Review timeline:	Submission date:	14 January 2014
	Editorial Decision:	06 February 2014
	Revision received:	21 July 2014
	Editorial Decision:	12 August 2014
	Revision received:	09 September 2014
	Editorial Decision:	30 September 2014
	Revision received:	27 October 2014
	Accepted:	03 November 2014

Editor: Maria Polychronidou

Transaction Report:

1st Editorial Decision

06 February 2014

Thank you again for submitting your work to Molecular Systems Biology. We have now heard back from the three referees who agreed to evaluate your manuscript. As you will see from the reports below, the referees acknowledge that you present potentially interesting findings. However, they raise a series of concerns, which should be carefully addressed in a revision of the manuscript.

Without repeating all the points listed below, some of the more fundamental issues are the following:

- The advantages of the presented approach should be clearly demonstrated (points 3 and 5 of reviewer #1). In particular, the possibility that alternative explanations (other than the proposed increased statistical power of the method) could underlie the unusually high number of detected trans-eQTLs compared to cis-eQTLs needs to be addressed.
- Additional experimentation is required to better support the presented conclusions regarding the effects of the *swc5-fs* allele and to directly demonstrate that genetic variation of *swc5* compromises the deposition of H2A.Z.

On a more editorial level, and along the lines with comment #4 of reviewer #1, we would like to ask you to make all datasets available by depositing them in the appropriate databases or by including them in the Supplementary Information.

If you feel you can satisfactorily deal with these points and those listed by the referees, you may wish to submit a revised version of your manuscript. Please attach a covering letter giving details of the way in which you have handled each of the points raised by the referees. A revised manuscript

will be once again subject to review and you probably understand that we can give you no guarantee at this stage that the eventual outcome will be favorable.

Reviewer #1:

Comments for Clement-Ziza et al. paper

Using a cross of fission yeast Clementa-Ziza et al. performed mapping of quantitative trait loci underlying expression levels of both sense and antisense transcripts. Previous efforts to study eQTLs were mostly carried out in model organisms including *Saccharomyces cerevisiae*. To our knowledge, this is the first study to collect association data with a recombinant cross of *Schizosaccharomyces pombe*. I believe the constructed fission yeast cross together with its accompanying data (if made available) will provide valuable resource for future genetic studies. Although the experimental design and QTL mapping method lack sufficient novelty, the detailed experimental validation on one of the casual genes *swc5* offers new insight on how a QTL could modulate its target genes. Overall I recommend a major revision of the manuscript with specific comments below.

Major comments:

1. The title is not an accurate representation of the findings presented in this study. The title suggests regulation of transcription is mediated through antisense transcripts but there is no concrete evidence to support this in the text.

2. To demonstrate that the frame shift of *swc5* reduces H2A.Z deposition, ChIP-seq data for H2A.Z were compared in the parental strains. This is not strong evidence, since any other genetic variants in the parental strains could confound the observed differences of H2A.Z between the two strains. To attribute the differences in H2.Z marks to *swc5*-fs, the comparison should be performed with *swc*-del, *swc*-allele replacement or a pool of segregants containing the *swc*-fs against the wild type.

3. Alignment to strain specific genome: The authors claim that use of strain specific genomes improves transcript quantification. However, increase in the read counts of transcripts of interest is not evidence that the quality of the quantification is improved. Evidence should be made using simulated reads where the ground truth is known (e.g. Degner et al. 2009 show that by using strain specific genomes, more reads from the non-reference genome are mapped to the transcripts, without a compromise on mapping accuracy).

4. Release of data: The paper claimed in the abstract, "The strains, methods, and datasets generated here provide a rich resource for future studies" (similar claim in the end of discussion). However, the raw sequencing data of the strains (RNAseq, DNAseq and ChIP-seq) and the measured growth curves (raw files) were not provided in the manuscript. To serve as a useful resource, aforementioned raw data should be made available through either the supplementary data or deposition in a public database. Furthermore, mapping results and mapped eQTLs with their targets should also be provided.

5. Claims of statistical power: in the results section discussing the increased number of trans-eQTLs versus cis-eQTLs, the authors strongly concluded that this is due to their mapping method and the design of the study ("we attribute this discrepancy to the increased statistical power of our approach"). There are other equally possible factors:

1) There are only 708 markers and a cis-eQTL is defined as a QTL within 10 kb window of the target transcript. For each transcript, there is only a limited number of possible cis-QTLs, hence you would expect a reduction in the number of cis-eQTLs. For trans-eQTLs, the small number of markers greatly reduces the burden of multiple testing. In comparison, human or mouse eQTL studies have more than 100 times the number of markers.

2) A significant portion of the trans-eQTLs comes from the *swc*-fs and also from chromosome 1

inversion. Diverse karyotypes have been observed in the *S. pombe* population which can seriously influence genetic architecture of *S. pombe* crosses, making it different from that of other model organisms (mouse, yeast, arabidopsis)

Some of these were discussed briefly in the discussion (" the absolute number of cis-eQTL that could be detected will largely be determined by the genetic similarity (or difference) of the strains being crossed.")

If indeed the larger number of QTLs is due to the applied methods, a comparison should be made between their method and the conventional eQTL mapping methods (spearman correlation with proper FDR controls) or using a simulation like in other studies (e.g. Stegle et al. 2010 or Leek et al. 2007) to demonstrate clearly the advantages of their method. The cited reference Michaelson and Beyer 2010 does not demonstrate an increase in statistical power to detect QTL.

If the power would come from deep coverage of RNAseq, the authors should demonstrate how the power of their method changes with depth, instead of citing Montgomery et al., which makes no such claims.

Minor comments:

Page 3: "pioneering work has demonstrated that RNA-seq increases the statistical power for detecting eQTL". Montgomery et al. 2010 was cited here to support the statement. However the cited paper does not claim to gain statistical power in the mapping of eQTL. Montgomery et al. demonstrated that more eQTLs were identified using exon quantification with RNAseq data. This implies an increase in resolution and not necessarily suggests statistical power for detecting more eQTLs. If one transcription unit is defined as a gene, like in arrays, the number of eQTLs in RNAseq is similar to that found in microarray studies

Page 3: "have only identified a few distant eQTLs". This is not an accurate statement since Battle et al. 2014 reported more than 200 eQTLs in their study, which is not a few.

Page 3: "Microarray based eQTL studies have shown that the vast majority of eQTLs act in trans". This is still debatable and has only been shown in budding yeast (Yvert et al. 2003) with sufficient samples. In other organisms including human and *C. elegans*, due to multiple testing burdens, how most eQTLs modulate the target genes needs to be investigated.

Page 3: "Further, available eQTL studies have not exploited the genotype information contained in the sequencing data". Genotyping using RNAseq data has been exploited in previous studies and thus is a natural step to take in eQTL study. The comparison to these studies cited is rather unfair, because genotyping with RNAseq is more difficult in diploid organisms. The challenges include size of haplotype blocks, allele specific expression, RNA editing.

Page 4: The manuscript does not report the results from crossing of different *S. Pombe* strains and cited another paper. This result might be useful for others who seek to try other crosses between *S. Pombe* strains

Page 5: "reliably measure the expression". To support this, the authors should provide the correlation coefficient or more convincingly showing scatterplots between the replicates.

Page 6: The average depth of the RNAseq or the length of the sequencing reads cannot be found in the result section as well as throughout the manuscript. It would be extremely useful to provide standard summary statistics for the sequencing libraries.

Page 7: An FDR cutoff of 5% was used for the detection of eQTLs. However the estimation of FDR was not elaborated in the methods. Tellingly, method should sufficiently describe the analysis and experimental procedures to enable others to reproduce the study. For authors' reference, a good example (also used random forest method to map trans-eQTL) can be in Francesconi and Lehner 2014.

Page 9: Regarding the eQTL hotspot detection, identification of hotspots in eQTL studies can be prone to artifacts due to correlated traits (Breitling et al. 2008). The authors should compare to other

methods such as the one discussed in Breitling et al 2008 to robustly detect the presence of an eQTL hotspot.

Page 10: The authors suggested that *swc5* could be casual gene by comparing the correlation coefficient of expression between *swc-del* strains and segregants containing different *swc5* alleles. Based on the difference in correlation coefficients (median of 0.93 versus 0.95 as determined from figure 5A) they presented this as strong evidence to support their hypothesis. A more direct approach to corroborate the hypothesis would be differential gene expression analysis. More specifically, the differential expressed genes between the *swc5* deletion and wild type (968 parental strain) should be the targets of the eQTLs in hotspot 11.

Page 12: For figure 6, the legends for C and D are swapped. It's also wrongly referred to in the text.

Page 21: It should be explained why 50kb was chosen as a limit to imputing genotypes. Theoretically from the recombination events, the average rate of recombination between 2 loci 50kb apart can be estimated. (According to previous classical genetic studies, 10kb \approx 1cM) So the probability of an error when imputing a genotype with agreeing flanking markers (same parental genotype) would be the probability of a double crossover, or non-crossover.

Other specific minor comments:

1. In other studies examining antisense transcription, actinomycin D was used to minimize the antisense artifacts caused by reverse transcriptase (Lardenois et al. 2011, Guisbert et al. 2012). One sanity check is to include in supplementary, a scatterplot of the expression levels of the antisense transcripts against the expression levels of their sense counter parts. If the correlation between sense and antisense is low, it would assure readers and other researchers that the aseQTLs are not artifacts.

2. No results have been presented for QTL mapping of different isoforms and/or alternative splicing. Nonetheless, the advantages of using RNAseq for these traits are constantly mentioned in the manuscript. The emphasis on this can be reduced.

References:

- Avelar, 2013. Genome architecture is a selectable trait that can be maintained by antagonistic pleiotropy. *Nature Commun.*
- Battle, 2014. Characterizing the genetic basis of transcriptome diversity through RNA-sequencing of 922 individuals. *Genome Research*
- Breitling, 2008. Genetical genomics: spotlight on QTL hotspots. *PLoS Genet*
- Guisbert, 2012. Meiosis-induced alterations in transcript architecture and noncoding RNA expression in *S. cerevisiae*. *RNA*
- Francescon, 2014. The effects of genetic variation on gene expression dynamics during development. *Nature*
- Michaelson, 2010. Data-driven assessment of eQTL mapping methods. *BMC Genomics*
- Lardenois. 2011. Execution of the meiotic noncoding RNA expression program and the onset of gametogenesis in yeast require the conserved exosome subunit Rrp6. *Proc Natl Acad Sci U S A*
- Leek, 2007. Capturing heterogeneity in gene expression studies by surrogate variable analysis. *PLoS Genet.*
- Stegle, 2010. A Bayesian framework to account for complex non-genetic factors in gene expression levels greatly increases power in eQTL studies. *PLoS Comput Biol.*
- Storey, 2003. Statistical significance for genomewide studies. *Proc Natl Acad Sci U S A*

Yvert, 2003. Trans-acting regulatory variation in *Saccharomyces cerevisiae* and the role of transcription factors. *Nature Genetics*

Reviewer #2:

The manuscript of Clement-Ziza et al. describes a QTL study in *S. pombe*. After generating and phenotyping 48 recombinant strains, the authors performed a strand-specific RNA-seq and developed a computational framework to identify cis- and trans-eQTLs. Among 11 hotspots over the genome, the authors present the striking example of a variation affecting antisense transcription. Further dissections of the mechanism indicate that the histone H2AZ chaperone Swr5 has been mutated. Genome-wide data show that H2A-Z density is indeed affected in the *swr5* mutant, suggesting that the reduction of H2A-Z participate in the antisense derepression.

The manuscript is well written, referenced and easy to read. Results are of general interest and present a very good example of how eQTL analysis can lead to the identification of coding (or non-coding) drivers for genome expression regulation. This manuscript warrants publication in MSB journal acknowledging that the authors would clarify if the method allows the finding of ncRNA drivers for trans-eQTL but also would adjust the last part of the results. Indeed, as it stands, it presents typos and poor experimental quality damaging the interpretation of this part. If (and only if) these 2 major critics are addressed, the manuscript could be considered for publication.

Major critics:

1-The last part of the results is very unclear and the mislabelling of the figures legends (see legend D for C etc..) somehow proves that this part is less finished than the rest of the manuscript. Several concerns can be raised:

-It is not clear how the authors define antisense as figure 6C show convergent genes with no overlap? Antisense should be convergent overlapping RNAs. What are the evidence for that in the examples taken here? What is the purpose of figure 6D, that now presents convergent (not overlapping?) genes?

- Figure 6C is everything but quantitative. Ratios calculation should be done using Real-time PCR approaches and not agarose gel quantification. This experiment is essential since it validates the RNA-seq but also the fact that H2A-Z mutant behave like *swr5* mutant in term of antisense expression.

-Cyp7 example should be discarded since it contains a true ncRNA antisense located in the first part of the genes (SPBC16H5-15). Maybe to avoid any confusion the authors should select other cases with true overlapping convergent genes.

2- It would have been very interesting to determine whether (long) non-coding RNAs are among the trans-eQTL drivers since it is today one of the major question in the field. eQTL offers one of the best strategy and argument towards understanding a function for those ncRNA. Could the authors list, at the locus resolution, the putative non coding regions that could act on global gene expression regulation? Is there any non-coding region that would be listed among the 11 hotspots described in Figure supplementary 14? Maybe the 11 hotspots could be refined into smaller bins to identify the coding or non-coding genes?

Reviewer #3:

This manuscript reports the mapping of genetic variants that control differences in expression of coding and antisense transcripts between *S. pombe* strains. The work is carefully done although I note one major issue below, and a number of minor suggestions.

Major:

The authors start with two distinct *S. pombe* isolates and carry out two rounds of crossing to generate 44 F2 lines, whose growth and expression they profile in standard medium. They use RNA-seq reads as input into a standard genotype caller to infer inheritance at each of 708 markers in each F2 strain. The authors then identify markers that show co-inheritance across strains with expression

of either mRNAs or noncoding RNAs. The authors focus on a frameshift mutation in the H2A.Z deposition factor *swc5* which links to hundreds of mRNA and antisense transcript levels. They show nicely that expression of linking transcripts is recapitulated by a *swc5* delete strain. They also carry out ChIP in the parental strains and show differences at the affected loci in H2A.Z occupancy, which suggests, though it doesn't prove, the reasonable model that the *swc5* effects on expression proceed through H2A.Z. It is thus not correct to say that the paper "identified a genetic variation of *swc5* that compromises the deposition of the histone variant H2A.Z." I'm happy to believe this a likely model (maybe even a no-brainer) but formally, the authors would need an allele replacement experiment to prove it.

Likewise, Figure 6a and b show the anticorrelation, across progeny of the cross, between mRNAs and their overlapping antisenses that show linkage to the *swc5* variant; in Figure 6c is there a well-controlled experiment on the *swc5* deletion and other chromatin remodeler mutants that validates their causal relationship with sense and antisense expression at a handful of loci. I disagree, however, that the authors have implicated read-through transcription as causal for regulatory effects on mRNAs. Figure 6c shows quantitative PCR on a given antisense locus and does not delineate the length of the transcript, whether it is truly a product of read-through. It also does not prove that the antisense controls sense expression; it is essentially another correlative analysis as in Figure 6a, just with different genetic backgrounds. The last section of the results overstates the conclusions possible from the data, in my view: "compromised H2A.Z deposition resulting from the loss of functional Swc5 in the *swc5*-fs strains leads to enhanced read-through transcription, which in turn negatively affects the expression of neighboring genes via antisense transcription." This again may be an appealing model but has not been proven.

Minor:

Genotyping from expression data is now routine and the authors should cite papers that have established this precedent (e.g. Piskol et al., *AJHG* 2013; Quinn et al., *PLoS ONE* 2013). The authors seem to be right that whole-genome linkage or association mapping of gene expression studies have not previously used amended genomes. However, analyses of allele-specific expression in heterozygote diploids by RNA-seq (to identify cis-eQTLs) have used this approach, and have already made the point that it affects expression estimates (e.g. Turro et al., *Genome Biology* 2011; Vijaya Satya et al., *NAR* 2012; Lee et al., *G3* 2013; Stevenson et al., *BMC Genomics* 2013; Vinay Pandey et al., *Molecular Ecology* 2013). Thus, it is not appropriate for the authors to say that "[t]his potential artifact in RNA-seq-based eQTL studies has not been investigated" (p. 7). Instead, the literature on amending genomes for RNA-seq mapping should be cited, with the authors' Figs S10-13 (and mention in the text) not necessary in their current expanded form. I will note that without QPCR or other independent validation we do not strictly know whether amending genomes improves accuracy of expression estimates.

The authors that a large fraction of transcripts show linkage to trans-acting variants, a slightly unusual pattern in a data set like this. I would guess that it is a product of the serendipitous presence in their parent strain pair of two loci, an inversion on chromosome I and the *swc5* locus they characterize (see below), that have very widespread effects on expression; it's not clear that this reveals any particular biological principle of relevance for the field.

I was surprised to see that other than the *sw5* locus, the authors say little about the biology of the mRNAs that vary between isolates or show linkage with markers, the Gene Ontology enrichment among mRNAs that show linkage to major trans-acting loci, their relationship to growth, etc. I acknowledge that it's been done in many systems already and that the antisense transcripts are what's new here, but it seems like an odd omission not to make some statements about the biological relevance of the bulk of their data in presenting Supplementary Table 2.

1st Revision - authors' response

21 July 2014

We would like to thank you for your consideration of our manuscript now entitled "Natural genetic variation impacts expression levels of coding, non-coding and antisense transcripts in fission yeast" by Clément-Ziza et al. Thank you for the exceptional extra time you granted us; it is our great

pleasure to submit a revised version of the manuscript to Molecular Systems Biology.

We were very impressed by the very high quality of the reviews and we seriously mean that: we rarely get reviews of similar quality. We thank all three Reviewers for the time they spend on working with our submission and we are convinced that the reviews have helped to drastically improve the paper. We have thoroughly addressed all issues raised by the Reviewers by performing additional experiments, analysis and by amending the manuscript.

In particular:

- we have carried out analysis of simulated RNA-seq data to assess the performance of our proposed method for RNA-seq based expression quantification,
- we further analyzed and discussed our result concerning the high proportion of trans-eQTL compared to cis-eQTL,
- we have performed additional well-controlled experiments (ChIP-seq and qPCR) that further support our conclusion regarding the effect the *swc5* frame shift polymorphism,
- we have included all data, which includes the raw, the intermediate and the final data. Altogether this constitutes an exceptionally rich set for other studies or method development.

Reviewer #1:

Comments for Clement-Ziza et al. paper

*Using a cross of fission yeast Clementa-Ziza et al. performed mapping of quantitative trait loci underlying expression levels of both sense and antisense transcripts. Previous efforts to study eQTLs were mostly carried out in model organisms including *Saccharomyces cerevisiae*. To our knowledge, this is the first study to collect association data with a recombinant cross of *Schizosaccharomyces pombe*. I believe the constructed fission yeast cross together with its accompanying data (if made available) will provide valuable resource for future genetic studies. Although the experimental design and QTL mapping method lack sufficient novelty, the detailed experimental validation on one of the casual genes *swc5* offers new insight on how a QTL could modulate its target genes. Overall I recommend a major revision of the manuscript with specific comments below.*

Major comments:

1. The title is not an accurate representation of the findings presented in this study. The title suggests regulation of transcription is mediated through antisense transcripts but there is no concrete evidence to support this in the text.

The point raised by Reviewer #1 is indeed pertinent. We amended the title in this revised version to: "Natural genetic variation impacts expression levels of coding, non-coding and antisense transcripts in fission yeast"

*2. To demonstrate that the frame shift of *swc5* reduces H2A.Z deposition, ChIP-seq data for H2A.Z were compared in the parental strains. This is not strong evidence, since any other genetic variants in the parental strains could confound the observed differences of H2A.Z between the two strains. To attribute the differences in H2.Z marks to *swc5*-fs, the comparison should be performed with *swc5*-del, *swc5*-allele replacement or a pool of segregants containing the *swc5*-fs against the wild type. We agree with Reviewer #1 in principle. The ChIP-seq analysis for H2A.Z in both parental strains is not sufficient to formally prove that *swc5*-fs is the molecular event responsible for the observed differences in H2A.Z deposition. This point has also been raised by Reviewer #3. As suggested by Reviewer #1, we conducted an additional set of ChIP-seq experiments with a *swc5*-deletion strain. To generate this strain, we crossed the 968-*pht1*-tagged strain that we previously generated with the *swc5*-deletion strain after having crossed out all auxotrophic markers. Two such clones were picked for the experiments and were checked by Western and PCR analysis (Figure E36). The H2A.Z occupancy profiles in *pht1*-tagged-*swc5* deletion strains recapitulated our previous observation made in Y0036, thus confirming that *swc5*-fs was the causal genetic variation for the reduced H2A.Z deposition at the +1 histone. These results were integrated in the manuscript (Figure 5d), the Results and Materials and Methods section corresponding to these analyses were updated (p.11-12 and p.21-22), as well as in Figure E36 showing the validation of the tagging in the *swc5*-del strain.*

3. Alignment to strain specific genome: The authors claim that use of strain specific genomes

improves transcript quantification. However, increase in the read counts of transcripts of interest is not evidence that the quality of the quantification is improved. Evidence should be made using simulated reads where the ground truth is known (e.g. Degner et al. 2009 show that by using strain specific genomes, more reads from the non-reference genome are mapped to the transcripts, without a compromise on mapping accuracy).

As suggested by Reviewer #1, we now performed simulations. We generated simulated RNA-seq reads for the two parental strains and three segregant strains. Note that sequencing errors were also simulated at a rate comparable to the one we observed in the real data, since Degner et al. showed that they were playing an important role in the expression quantification accuracy (see Materials and methods, Figure E14). Expression levels were then measured using both approaches (reference genome mapping and strain-specific genome mapping). Subsequently, the measurements were compared between them and with the ground truth. Results were similar to what we observed with the real data: expression measurements were higher when using strain-specific genomes. Moreover, comparison with the ground truth showed that strain-specific genome mapping was leading to improved quantification accuracy in the large majority of the concerned cases, although the magnitude of the gain was small. The Results and Methods sections of the manuscript (p.7-8, and p.27-28, respectively) as well as the Expanded View Material (Figure E14, E15, E16) were amended to present the result of these simulations.

4. Release of data: The paper claimed in the abstract, "The strains, methods, and datasets generated here provide a rich resource for future studies" (similar claim in the end of discussion). However, the raw sequencing data of the strains (RNAseq, DNaseq and ChIP-seq) and the measured growth curves (raw files) were not provided in the manuscript. To serve as a useful resource, aforementioned raw PRJEB6706database. Furthermore, mapping results and mapped eQTLs with their targets should also be provided.

The raw data of the growth curve measured with the BioLector microfermentation system are now provided as Dataset E1. Note that processed growth data (extracted growth parameters) are also provided (Dataset E2). The raw data of other survival assays at 36h are provided as Dataset E3. Raw DNA resequencing data were archived in the European Nucleotide Archive (www.ebi.ac.uk/ena/) under the accession numbers [ERX007392](http://www.ebi.ac.uk/ena/entry/ERX007392) and [ERX007395](http://www.ebi.ac.uk/ena/entry/ERX007395) for the strains 768 and Y0036 respectively. Raw RNA-seq data were deposited in ArrayExpress (<https://www.ebi.ac.uk/arrayexpress/>) under the accession number: E-MTAB-2640. Raw ChIP-seq data were also deposited in ArrayExpress under the accession number: E-MTAB-2650. These data will be publicly released upon acceptance of the article. The Material and Methods section have been updated to include the references to the raw data.

We provide here login information that enables the reviewers to access the ArrayExpress experiments, before their public release, for the reviewing procedure. Details about the login procedure can be found here http://www.ebi.ac.uk/arrayexpress/help/how_to_search.html#Login

Username: Reviewer_E-MTAB-2650

Password: efjmmtxk

Username: Reviewer_E-MTAB-2640

Password: h7kafbaa

We also included as Dataset, QTL mapping results as tables indicating for every trait the q-value of the linkage with all markers (separate tables for sense, antisense, and growth QTLs; Datasets E10, E12, and E14). Additionally we added a Supplementary Data Table with the genotypes and positions of all markers (Datasets E7 and E9). For the reader's convenience, we also provide tables of eQTL linkage and aseQTL linkage at 5%FDR (Datasets E11 and E13). For each association we indicated the target gene, the genomic coordinates of the QTLs, and whether it is a *cis* or *trans* QTL. Altogether, the raw data, the intermediate data, and the final result data are provided with this manuscript. This certainly constitutes a rich set that will be useful for other studies or method development. Note that all the provided data tables contain an explanatory comment header with a precise description of the content of each column.

Here is a complete list of the Datasets:

- Dataset E1: raw growth data
- Dataset E2: extracted growth traits
- Dataset E3: survival data
- Dataset E4: raw sense read counts
- Dataset E5: raw antisense read counts

- Dataset E6: detected genomic variation in the parental strains
- Dataset E7: genotype data for every variants for the entire library
- Dataset E8: recombination position for every strains of the library
- Dataset E9: genotype data used for the mapping with the mapping marker position
- Dataset E10: sense eQTL mapping results (q-value of linkage for every trait and marker)
- Dataset E11: list of the significant eQTL with their position, target and putative regulator
- Dataset E12: aseQTL mapping results (q-value of linkage for every trait and marker)
- Dataset E13: list of the significant aseQTL with their position and target
- Dataset E14: growth QTL results
- Dataset E15: list of eQTL that do not contain any coding gene
- Dataset E16: raw qPCR data
- Dataset E17: proteomics skylines and transitions

5. *Claims of statistical power: in the results section discussing the increased number of trans-eQTLs versus cis-eQTLs, the authors strongly concluded that this is due to their mapping method and the design of the study ("we attribute this discrepancy to the increased statistical power of our approach"). There are other equally possible factors:*

1) *There are only 708 markers and a cis-eQTL is defined as a QTL within 10 kb window of the target transcript. For each transcript, there is only a limited number of possible cis-QTLs, hence you would expect a reduction in the number of cis-eQTLs. For trans-eQTLs, the small number of markers greatly reduces the burden of multiple testing. In comparison, human or mouse eQTL studies have more than 100 times the number of markers.*

Our wording in the MS was somewhat misleading: with “our approach” we did not intend to refer only to our mapping method, but to the study design in general. We completely agree with the reviewer: the relatively small number of polymorphisms between the strains increases the statistical power for detecting QTLs. The number of cis-eQTLs is limited and thus, increasing the statistical power will eventually saturate the number of cis-QTLs that could be detected, while the detectable trans-eQTLs may continue to rise.

In reading the remark of Reviewer #1 and the way we described the definition of cis- versus trans-QTL in the methods section of the manuscript, we realized that we did not explain clearly enough our method for this point. This has evidently led to a misunderstanding. In our study, each of the 708 mapping markers is a set of polymorphisms in complete linkage disequilibrium. Therefore each marker represents a genomic interval located in between the most 5' and 3' polymorphisms in LD. However, when we found a linkage between a marker and a trait, we considered the QTL as being a larger locus defined as in between the two flanking markers as follows: the genomic interval located between the “end” of the LD region of the previous marker and the “start” of the LD region of the next markers. Therefore the QTLs overlap. Linkages with last (or first) marker at the end (or beginning) of a chromosome were considered as enclosing the entire telomere. QTL linkage was considered to act in *cis* when any boundaries of the target gene were located at less than 10 kb of the genomic interval considered as a QTL. We consider this definition of *cis* QTL as permissive. Each linkage has more than 1/708 chances to be considered as *cis*. This is actually much more than in other organisms with a bigger genome. We amended the manuscript to describe in details the delimitation of the QTLs in the method section in the *QTL mapping* section (p.30), and the definition of the *cis* linkage *cis versus trans QTL* section (p.31).

This does not affect the other remarks of Reviewer #1 about the reduced burden of multiple testing compared to human or mouse eQTL. However, we consider this as a gain of statistical power to detect QTL due to small size of the *S.pombe* genome.

2) *A significant portion of the trans-eQTLs comes from the swc-fs and also from chromosome 1 inversion. Diverse karyotypes have been observed in the S. pombe population which can seriously influence genetic architecture of S. pombe crosses, making it different from that of other model organisms (mouse, yeast, arabidopsis)*

We agree with Reviewer #1 that the particularity of the cross, in particular the chromosome 1 inversion and the *swc5* hotspot, can inflate the number of trans-eQTL. This point has also been noticed by Reviewer #3. To test this, we repeated the *cis* versus *trans* analysis excluding the loci corresponding to the hotspots (in particular hotspot 11 corresponding to *swc5*-frame shift) and the entire chromosome I inversion. This indeed led to an increase of the *cis*-QTL proportion (6.8% of *cis*-eQTL among all eQTL). However, this still remains much lower than what has been published

and observed before (to our knowledge). This point and analysis have been added in the manuscript in the Discussion (p.15).

Some of these were discussed briefly in the discussion (" the absolute number of cis-eQTL that could be detected will largely be determined by the genetic similarity (or difference) of the strains being crossed.")

If indeed the larger number of QTLs is due to the applied methods, a comparison should be made between their method and the conventional eQTL mapping methods (spearman correlation with proper FDR controls) or using a simulation like in other studies (e.g. Stegle et al. 2010 or Leek et al. 2007) to demonstrate clearly the advantages of their method. The cited reference Michaelson and Beyer 2010 does not demonstrate an increase in statistical power to detect QTL.

If the power would come from deep coverage of RNAseq, the authors should demonstrate how the power of their method changes with depth, instead of citing Montgomery et al., which makes no such claims.

Altogether, we completely agree with Reviewer #1. We actually did *not* want to claim that the low proportion of cis-eQTL was only due to a gain of statistical power resulting of the applied methods. We only consider this as a possibility. Moreover, we do not believe that the gain of power to detect QTLs is exclusively due to the method applied. The small genome and the genetic similarity of the parental strains reduced the trait complexity (i.e. number of polymorphisms affecting a trait) compared to other studies, which increased the statistical power for detecting QTL. The methods (experimental and analytic) may play an additional role. We amended the manuscript to make this point clear everywhere where we spotted a possible misunderstanding. The Discussion was expanded to better describe the possible explanations.

Minor comments:

Page 3: "pioneering work has demonstrated that RNA-seq increases the statistical power for detecting eQTL". Montgomery et al. 2010 was cited here to support the statement. However the cited paper does not claim to gain statistical power in the mapping of eQTL. Montgomery et al. demonstrated that more eQTLs were identified using exon quantification with RNAseq data. This implies an increase in resolution and not necessarily suggests statistical power for detecting more eQTLs. If one transcription unit is defined as a gene, like in arrays, the number of eQTLs in RNAseq is similar to that found in microarray studies

The manuscript has been amended accordingly, and the sentence was removed.

Page 3: "have only identified a few distant eQTLs". This is not an accurate statement since Battle et al. 2014 reported more than 200 eQTLs in their study, which is not a few.

We agree with Reviewer #1: Battle et al. have identified 269 intra-chromosomal and 138 inter-chromosomal eQTLs, which together make 407 trans-eQTLs (Table 1, Battle et al, 2014, PMID: 24092820). However, these numbers are relatively small (4.2%) compared to the total number of eQTL they detected (10,914 cis-eQTLs were mapped). Therefore we amended the manuscript and the sentence was replaced by: "have identified only a relatively small proportion of distant eQTLs".

Page 3: "Microarray based eQTL studies have shown that the vast majority of eQTLs act in trans". This is still debatable and has only been shown in budding yeast (Yvert et al. 2003) with sufficient samples. In other organisms including human and C. elegans, due to multiple testing burdens, how most eQTLs modulate the target genes needs to be investigated

On this point, we disagree with Reviewer 1: Using mouse eQTL data, we have previously shown that increasing statistical power increases the ratio of trans-versus-cis-eQTL (i.e. the fraction of trans-eQTL increases). We could show that with sufficient statistical power one can detect more trans- than cis-eQTL even in mice (Ackermann et al, 2013, PMID: 23754949), which is in agreement with the other references we cite. However, we have rephrased this sentence and updated the references after this sentence: "It has been suggested that that the vast majority of eQTLs act in trans (Ackermann et al, 2013; Brem et al, 2002; Yvert et al, 2003; Rockman & Kruglyak, 2006)"

Page 3: "Further, available eQTL studies have not exploited the genotype information contained in

the sequencing data". Genotyping using RNAseq data has been exploited in previous studies and thus is a natural step to take in eQTL study. The comparison to these studies cited is rather unfair, because genotyping with RNAseq is more difficult in diploid organisms. The challenges include size of haplotype blocks, allele specific expression, RNA editing.

We agree that RNA-seq based genotyping would be more complicated in diploid (heterozygous) organisms. We therefore amended the manuscript and presented in a more fair way the RNA-seq based genotyping, as following on page 3: "Further, although the sequence variation information contained in the RNA-seq data has been exploited for genotyping (for instance: Quinn et al, 2013), it has not been used in the framework of eQTL mapping, possibly due to the complexity of the studied organisms (Keane et al, 2011; Montgomery et al, 2010, 2011; Pickrell et al, 2010)."

Page 4: The manuscript does not report the results from crossing of different S. Pombe strains and cited another paper. This result might be useful for others who seek to try other crosses between S. Pombe strains

These results will be published in a separate publication already submitted. It has been referenced as *manuscript in preparation*, as required by Molecular Systems Biology. Reference to it will be: Daniel C Jeffares, Charalampos Rallis, Adrien Rieux, Doug Speed, Martin Převorovský, Tobias Mourier, Francisc X Marsellach, Zamin Iqbal, Winston Lau, Tammy Cheng, Rodrigo Pracana, Michael Muelleder, Jonathan Lawson, Anatole Chessel, Sendu Bala, Garrett Hellenthal, Brendan O'Fallon, Thomas Keane, Jared T Simpson, Leanne Bischof, Bartłomiej Tomiczek, Danny Bitton, Theodora Sideri, Sandra Codlin, Josephine EEU Hellberg, Laurent van Trig, Linda Jeffery, Sophie Atkinson, Malte Thodberg, Melanie Febrer, Kirsten McLay, Nizar Drou, William Brown, Jacqueline Hayles, Rafael E Carazo Salas, Markus Ralser, Nikolas Maniatis, David J Balding, Francois Balloux, Richard Durbin, Jürg Bähler . The Genomic and Phenotypic Diversity of Schizosaccharomyces pombe.

Page 5: "reliably measure the expression". To support this, the authors should provide the correlation coefficient or more convincingly showing scatterplots between the replicates.

As suggested by Reviewer #1, we provided the correlation coefficient between the replicates and showed scatter plots (see Figure E7). We also amended the text in the manuscript and added the following sentence (page 5): "The average Pearson's correlation coefficient between the replicates was 0.971 (Figure E7)."

Page 6: The average depth of the RNAseq or the length of the sequencing reads cannot be found in the result section as well as throughout the manuscript. It would be extremely useful to provide standard summary statistics for the sequencing libraries.

As suggested by Reviewer #1, we added information about sequencing. Average RNA-seq statistics were provided in Table 1 and detailed ones in Table E2. Moreover, the length of the RNA-seq reads (50 bases) is mentioned in the Results and Methods sections (pages 5 and 21).

Page 7: An FDR cutoff of 5% was used for the detection of eQTLs. However the estimation of FDR was not elaborated in the methods. Tellingly, method should sufficiently describe the analysis and experimental procedures to enable others to reproduce the study. For authors' reference, a good example (also used random forest method to map trans-eQTL) can be in Francesconi and Lehner 2014.

We agree that our previous description of the eQTL mapping method was incomplete. We therefore amended the manuscript to fully describe it (p.29-30).

Page 9: Regarding the eQTL hotspot detection, identification of hotspots in eQTL studies can be prone to artifacts due to correlated traits (Breitling et al. 2008). The authors should compare to other methods such as the one discussed in Breitling et al 2008 to robustly detect the presence of an eQTL hotspot.

As Reviewer #1 mentioned, Breitling et al. have shown that the permutation strategy used to assess the statistical significance of the QTL linkage can lead to an inflation of the number of hotspots. In their work, they propose to permute the strain labels, so that each strain is assigned the genotype vector of another random strain, while the expression matrix is unchanged (*sic*). The goal of this

permutation strategy is to keep the correlation structure of the gene expression. When we mapped the QTL in this study, we took this issue into account. We used a slightly different permutation strategy that is equivalent in regard of the issue raised by Breitling et al. Each trait vector was permuted and the Random Forest was then grown, keeping the genotype matrix unchanged. However, the same permutation scheme was used for every trait in order to keep the correlation between traits, which is the point that Breitling et al. advocated as important for hotspot detection. This procedure is equivalent to Breitling et al.'s strategy. Moreover, p-values were computed by estimating the null-distributions on a marker specific manner. This point was obviously not sufficiently well explained in the previous version of the manuscript. We therefore amended the Results and Methods sections (pages 10 and 30, 2nd §, respectively) to make it clear and cited the work of Breitling et al.

Page 10: The authors suggested that swc5 could be casual gene by comparing the correlation coefficient of expression between swc-del strains and segregants containing different swc5 alleles. Based on the difference in correlation coefficients (median of 0.93 versus 0.95 as determined from figure 5A) they presented this as strong evidence to support their hypothesis. A more direct approach to corroborate the hypothesis would be differential gene expression analysis. More specifically, the differentially expressed genes between the swc5 deletion and wild type (968 parental strain) should be the targets of the eQTLs in hotspot 11.

We thank Reviewer #1 for this suggestion. We performed a RNA-seq based differential gene expression analysis of $\Delta swc5$ (measured in triplicate) and the wild-type 968 strain (measured in septuplate in the eQTL dataset). We found 1,752 gene and 612 anti-sense traits differentially expressed between $\Delta swc5$ and the parental 968 strain at 5% FDR. We indeed found that the differentially expressed genes between the *swc5*-deletion and the parental 968 strain were significantly overlapping with the targets of the hotspot 11 ($P < 1e-58$ and $P < 1e-13$, respectively for the sense and antisense traits). Differential gene expression analysis was carried out using DeSeq2 (doi:10.1186/gb-2010-11-10-r106). Although highly significant, the overlap was not perfect, suggesting that other genetic factors may play a role. This further confirms that *swc5-fs* is implicated in the hotspot 11. The manuscript was amended to include this analysis (see Results page 11, Method page 29 and Figure E26)

Page 12: For figure 6, the legends for C and D are swapped. It's also wrongly referred to in the text. The content of the old Figure 6 has been completely updated and reorganized in the manuscript. This problem does not apply anymore.

Page 21: It should be explained why 50kb was chosen as a limit to imputing genotypes. Theoretically from the recombination events, the average rate of recombination between 2 loci 50kb apart can be estimated. (According to previous classical genetic studies, 10kb \approx 1cM) So the probability of an error when imputing a genotype with agreeing flanking markers (same parental genotype) would be the probability of a double crossover, or non-crossover.

We amended the manuscript to show that the limit of 50kb to input the genotype was justified. As Reviewer #1 suggested, we computed the probability (using a binomial model) of observing two recombinations in a 50kb interval considering that there were up to 100 recombination events in every segregant. This probability was small (0.0078). Note that 100 recombinations should be a large over estimation (even for F2), since it has been reported that there are approximately 50 crossovers per meiosis in *S.pombe* (Munz et al. 1989). Further, not all crossovers result in a recombination when only one segregant of the tetrad is considered. The section of the Expanded View Text, in which the accuracy of the genotyping is discussed, has been amended to include this explanation (see Expanded View Text page 2) and was referred to in the Methods section.

Other specific minor comments:

1. In other studies examining antisense transcription, actinomycin D was used to minimize the antisense artifacts caused by reverse transcriptase (Lardenosis et al. 2011, Guisbert et al. 2012). One sanity check is to include in supplementary, a scatterplot of the expression levels of the antisense transcripts against the expression levels of their sense counter parts. If the correlation

between sense and antisense is low, it would assure readers and other researchers that the aseQTLs are not artifacts.

Indeed it has been shown by Perocchi et al. (NAR, 2007, PMID: 17897965) that the reverse transcription was a source of spurious increase of antisense expression. They have shown that they could lead to artefactual positive correlation of sense and antisense levels. In our experiments, we did not use actinomycin D to minimize this effect. However, we do not observe correlation between sense and anti-sense as shown in Perrochi et al. We rather observe the contrary with a significant anti-correlation of sense and antisense expression in accordance with the most common *cis* effects (self-regulatory circuits) of antisense transcript (Plechano&Steinmetz, 2013, PMID 24217315). We included a scatterplot of the expression levels of the antisense transcripts against the expression levels of their sense counterpart (Figure E34), as suggested by Reviewer #1.

2. No results have been presented for QTL mapping of different isoforms and/or alternative splicing. Nonetheless, the advantages of using RNAseq for these traits are constantly mentioned in the manuscript. The emphasis on this can be reduced.

The advantages of using RNAseq for these traits are now only mentioned in the Introduction (page 3) to present the previous work of other on RNA-seq based QTL and at the end of the Discussion (page 18) as an opening to what could be done with the dataset that is not yet covered by this manuscript.

References:

- Avelar, 2013. Genome architecture is a selectable trait that can be maintained by antagonistic pleiotropy. *Nature Commun.*
- Battle, 2014. Characterizing the genetic basis of transcriptome diversity through RNA-sequencing of 922 individuals. *Genome Research*
- Breitling, 2008. Genetical genomics: spotlight on QTL hotspots. *PLoS Genet*
- Guisbert, 2012. Meiosis-induced alterations in transcript architecture and noncoding RNA expression in *S. cerevisiae*. *RNA*
- Francescon, 2014. The effects of genetic variation on gene expression dynamics during development. *Nature*
- Michaelson, 2010. Data-driven assessment of eQTL mapping methods. *BMC Genomics*
- Lardenois, 2011. Execution of the meiotic noncoding RNA expression program and the onset of gametogenesis in yeast require the conserved exosome subunit Rrp6. *Proc Natl Acad Sci U S A*
- Leek, 2007. Capturing heterogeneity in gene expression studies by surrogate variable analysis. *PLoS Genet.*
- Stegle, 2010. A Bayesian framework to account for complex non-genetic factors in gene expression levels greatly increases power in eQTL studies. *PLoS Comput Biol.*
- Storey, 2003. Statistical significance for genomewide studies. *Proc Natl Acad Sci U S A*
- Yvert, 2003. Trans-acting regulatory variation in *Saccharomyces cerevisiae* and the role of transcription factors. *Nature Genetics*

Reviewer #2:

Remarks to the Author:

*The manuscript of Clement-Ziza et al. describes a QTL study in *S. pombe*. After generating and phenotyping 48 recombinant strains, the authors performed a strand-specific RNA-seq and developed a computational framework to identify cis- and trans-eQTLs. Among 11 hotspots over the genome, the authors present the striking example of a variation affecting antisense transcription. Further dissections of the mechanism indicate that the histone H2AZ chaperone Swr5 has been mutated. Genome-wide data show that H2A-Z density is indeed affected in the swr5 mutant, suggesting that the reduction of H2A-Z participate in the antisense derepression.*

The manuscript is well written, referenced and easy to read. Results are of general interest and present a very good example of how eQTL analysis can lead to the identification of coding (or non-coding) drivers for genome expression regulation. This manuscript warrants publication in MSB journal acknowledging that the authors would clarify if the method allows the finding of ncRNA drivers for trans-eQTL but also would adjust the last part of the results. Indeed, as it stands, it presents typos and poor experimental quality damaging the interpretation of this part. If (and only if) these 2 major critics are addressed, the manuscript could be considered for publication.

Major critics:

1-The last part of the results is very unclear and the mislabelling of the figures legends (see legend D for C etc..) somehow proves that this part is less finished than the rest of the manuscript. Several concerns can be raised:

-It is not clear how the authors define antisense as figure 6C show convergent genes with no overlap? Antisense should be convergent overlapping RNAs. What are the evidence for that in the examples taken here? What is the purpose of figure 6D, that now presents convergent (not overlapping?) genes?

We agree with Reviewer #2 that this part was certainly not clear enough. Antisense transcripts can originate from diverse sources: overlapping genes (convergent and divergent), bidirectional transcription, and read-through transcripts. Read-through antisense transcripts are due to improper termination of the transcription in genes organized convergently (the read-through transcript of a gene is then the antisense of gene located on the opposite strand in the convergent pair). See the Figure below. The orange and the blue genes are organized convergently; they do not overlap. The green transcript would correspond to the read-through transcript of the blue gene, which failed to terminate at the annotated end of the blue gene. This transcript will be antisense with respect to the orange gene. This phenomenon has been reported in the literature (see for instance Anver *et al*, 2014; Zofall *et al*, 2009; Gullerova & Proudfoot, 2008; Ni *et al*, 2010, detailed reference in the manuscript). Note that while read-through may affect any gene it only causes antisense transcription in the case of convergently organized genes. For instance, read-through at genes organized in tandem may cause transcriptional interference (PMID: 15922833).

Increased antisense levels caused by transcriptional read-through has been observed in a strain lacking H2A.Z (pht1, Zofall et al. 2009). In this study, we did not rely on existing annotation to define antisense transcription, but took into account all the RNA-seq reads falling on the opposite strand of coding genes (only the CDS part of the gene). We found antisense signals for genes with no known antisense gene, and no overlap with coding genes on the opposite strand. Moreover, we found QTL mapping to these signals, thus demonstrating that they are not noise.

Finally, we have conducted RACE experiments (FigureE31) in which we present evidence for a read-through transcript in strains with direct or indirect H2A.Z reduced occupancy.

This part of the manuscript has been substantially modified. Now, in the analysis of the gene pair organization we distinguish the cases of overlapping and non-overlapping genes. However, as stated in the Expanded View Text, 5' termination sites are uncertain in *S. pombe*, "Indeed, transcription

termination (and polyadenylation) sites are highly variable in fission yeast (Schlackow *et al*, 2013; Mata, 2013; Gullerova & Proudfoot, 2008; Gullerova *et al*, 2011)". Moreover, Figure 8 (ex 6d) now presents schemes of the gene pairs with antisense RNAs. In the results section (page 13), we also briefly present the origins of antisense transcription. Finally, the analysis of the gene pair arrangement is more detailed in the Expanded View Text (p.6-7). These changes have made this section more clear.

- Figure 6C is everything but quantitative. Ratios calculation should be done using Real-time PCR approaches and not agarose gel quantification. This experiment is essential since it validates the RNA-seq but also the fact that H2A-Z mutant behave like *swr5* mutant in term of antisense expression.

We now replaced the former Figure 6C, as suggested by Reviewer #2, by a figure showing the results from well-controlled strand specific qPCR measurements. Experiments have been performed with many controls and repeats, and taking extra caution to limit the synthesis of primer independent cDNA, which has been shown as a potential source of bias in antisense quantification (see manuscript). Results are presented in the manuscript (Results section p.13-14, Figure 7). They are more extensively analyzed in the Expanded View Text (p.8-9 and Figure E29). Precise experimental design, experimental and analytic methods are presented in the Materials and method section (p.23-24) and in Figure E28. Raw qPCR data are provided in Dataset E16.

-*Cyp7* example should be discarded since it contains a true ncRNA antisense located in the first part of the genes (*SPBC16H5-15*). Maybe to avoid any confusion the authors should select other cases with true overlapping convergent genes.

We thank the reviewer for this remark. We removed *cyp7* from the genes analyzed in detail. As discussed above, our analyses are not restricted to overlapping genes. However, we have included *cbd4* in the genes we tested by qPCR. *Cbd4* completely overlaps with the UTR of *thi4*, based on the current annotation (<http://www.pombase.org/spombe/result/SPAC23H4.09/>). Additional analysis has been conducted for this case showing that the variation in antisense expression is not only due to the overlap but to other mechanisms (see Expanded View Text p. 9 and Figure E29).

2- It would have been very interesting to determine whether (long) non-coding RNAs are among the *trans-eQTL* drivers since it is today one of the major question in the field. *eQTL* offers one of the best strategy and argument towards understanding a function for those ncRNA. Could the authors list, at the locus resolution, the putative non coding regions that could act on global gene expression regulation?

Fine-mapping of QTL and the identification of QTL regulators (drivers) are a difficult task (Bing & Hoeschele, 2005; Kulp & Jagalur, 2006; Suthram *et al*, 2008; Verbeke *et al*, 2013, detailed references in the Manuscript). We propose a prediction of the strongest putative regulator candidate based on correlation of the expression levels of the candidate regulators and the target QTL gene. Here, 708 markers were assembled covering genomic intervals in which several polymorphisms are in complete linkage disequilibrium in the cross. Since the genome of *S. pombe* is extremely dense (~6500 annotated coding and non-coding genes for 12.4Mb), the loci generally contain several coding or non-coding genes. Moreover the linkages can span over several loci. Still, as indicated in the manuscript on page 9, 22.8% of the strongest regulator candidates were ncRNAs.

To help readers interested in a deeper exploration of the regulators, we added Supplementary Data Sets. We provide Tables of *eQTL* linkages and *aseQTL* linkages at 5% FDR and indicate for each association the target genes, the genomic coordinates of the QTLs, and whether it is a *cis* or *trans* QTL (Datasets E11 and E13). For the sense QTL (Dataset E11), we also incorporate in this table our prediction of the strongest candidate regulator and an indication whether the regulator is a coding or non-coding gene.

To go further, we looked specifically at linkages for which we were almost certain that non-coding polymorphisms or ncRNA were implicated: these are QTL containing either no genes, or only non-coding genes. This list of 18 QTLs is also provided (Dataset E15). Note that we found that the strongest candidate regulator of hotspot 11 was *swc5*, which suggests that our approach is somehow informative.

The manuscript has been amended to present these sets (page 9)

Is there any non-coding region that would be listed among the 11 hotspots described in Figure supplementary 14? Maybe the 11 hotspots could be refined into smaller bins to identify the coding

or non-coding genes?

As explained above, the main issue to surely identify non-coding regulator genes is the size of the QTL. Refining the bins to define the hotspot will not help in refining the genetic loci. One would need to study more individuals to have more recombinations in the QTL region and therefore reduce the size of the QTL.

To provide the reader more information about the potential regulators of the hotspot, we predicted the most probable regulator of each hotspot based on a “voting” method. For each hotspot, we predicted the strongest regulator of each hotspot target (as previously, based on the highest correlation) and considered that the most frequently selected candidate regulator (coding or non-coding) as the regulator of the hotspot. This analysis has been added to the manuscript with the eQTL hotspot analysis in the results and methods (page 9-10 and 31 respectively), and the results were integrated in Table E3.

Reviewer #3:

*This manuscript reports the mapping of genetic variants that control differences in expression of coding and antisense transcripts between *S. pombe* strains. The work is carefully done although I note one major issue below, and a number of minor suggestions.*

Major:

*The authors start with two distinct *S. pombe* isolates and carry out two rounds of crossing to generate 44 F2 lines, whose growth and expression they profile in standard medium. They use RNA-seq reads as input into a standard genotype caller to infer inheritance at each of 708 markers in each F2 strain. The authors then identify markers that show co-inheritance across strains with expression of either mRNAs or noncoding RNAs. The authors focus on a frameshift mutation in the H2A.Z deposition factor *swc5* which links to hundreds of mRNA and antisense transcript levels. They show nicely that expression of linking transcripts is recapitulated by a *swc5* delete strain. They also carry out ChIP in the parental strains and show differences at the affected loci in H2A.Z occupancy, which suggests, though it doesn't prove, the reasonable model that the *swc5* effects on expression proceed through H2A.Z. It is thus not correct to say that the paper "identified a genetic variation of *swc5* that compromises the deposition of the histone variant H2A.Z." I'm happy to believe this a likely model (maybe even a no-brainer) but formally, the authors would need an allele replacement experiment to prove it.*

We agree with Reviewer #3. The ChIP-seq analysis for H2A.Z in both parental strains alone was not sufficient to formally prove that *swc5*-fs is the molecular event responsible for the observed differences in H2A.Z deposition. This point has also been raised by Reviewer #1. We performed an analysis of the H2A.Z deposition in a *swc5*-deletion strain, as suggested by Reviewer #1. The H2A.Z occupancy profiles in the *swc5*-del strains recapitulated the observation we made with Y0036. These results were integrated in the manuscript (Figure 5d), and the Results and Method section corresponding to these analyses were updated (p.11-12 and p.21-22), as well as Figure E36. Further, we have toned down the conclusion in the Abstract as follows: “We identified a genetic variation of *swc5* that modifies the levels of more than 1,000 RNAs, with effects on both sense and antisense transcription, and show that this effect most likely goes through a compromised deposition of the histone variant H2A.Z.”

*Likewise, Figure 6a and b show the anticorrelation, across progeny of the cross, between mRNAs and their overlapping antisenses that show linkage to the *swc5* variant; in Figure 6c is there a well-controlled experiment on the *swc5* deletion and other chromatin remodeler mutants that validates their causal relationship with sense and antisense expression at a handful of loci. I disagree, however, that the authors have implicated read-through transcription as causal for regulatory effects on mRNAs. Figure 6c shows quantitative PCR on a given antisense locus and does not delineate the length of the transcript, whether it is truly a product of read-through. It also does not prove that the antisense controls sense expression; it is essentially another correlative analysis as in Figure 6a, just with different genetic backgrounds. The last section of the results overstates the conclusions possible from the data, in my view: "compromised H2A.Z deposition resulting from the*

loss of functional *Swc5* in the *swc5-fs* strains leads to enhanced read-through transcription, which in turn negatively affects the expression of neighboring genes via antisense transcription." This again may be an appealing model but has not been proven.

We agree with Reviewer #3. We do not formally prove that read-through transcription is causal in this case. We amended the manuscript in the Results and Discussion section to make this point very clear. However, we consider this as a very likely hypothesis and discuss it extensively in the manuscript. Moreover, we are adding in this revised version of the manuscript additional evidence, yet not formally conclusive, that strength our hypothesis:

- First, we show via qPCR that the effect on gene expression that we observed in the gene linked to the hotspot 11 are also observed in deletions strains, (notably $\Delta pht1$ and $\Delta pht1\Delta clr4$) for which it has been established that read-through was at the origin of the increase in antisense expression (Zofall *et al.*, 2009; Ni *et al.*, 2010; Anver *et al.*, 2014, detailed references in the manuscript). However, we could not directly confirm (nor disprove) by qPCR the presence of read-through RNAs (because of sensitivity limits, see Expanded View Text, p.8-9).
- We performed 3'RACE for one pair of convergent genes, which suggest that read-through transcripts are more abundant in the parental strains carrying the *swc5-fs* and in several strains (notably $\Delta swc5$ and $\Delta pht1$) deleted for the genes involved in the suppression of read-through antisense RNAs (Figure E31).

Minor:

Genotyping from expression data is now routine and the authors should cite papers that have established this precedent (e.g. Piskol et al., AJHG 2013; Quinn et al., PLoS ONE 2013). The authors seem to be right that whole-genome linkage or association mapping of gene expression studies have not previously used amended genomes. However, analyses of allele-specific expression in heterozygote diploids by RNA-seq (to identify cis-eQTLs) have used this approach, and have already made the point that it affects expression estimates (e.g. Turro et al., Genome Biology 2011; Vijaya Satya et al., NAR 2012; Lee et al., G3 2013; Stevenson et al., BMC Genomics 2013; Vinay Pandey et al., Molecular Ecology 2013). Thus, it is not appropriate for the authors to say that "[t]his potential artifact in RNA-seq-based eQTL studies has not been investigated" (p. 7). Instead, the literature on amending genomes for RNA-seq mapping should be cited, with the authors' Figs S10-13 (and mention in the text) not necessary in their current expanded form. I will note that without QPCR or other independent validation we do not strictly know whether amending genomes improves accuracy of expression estimates.

We agree that detecting genomic variation via RNA-seq has already been proposed. However, we proposed something slightly different since the RNA-seq is not used to detect the variation but to infer the genotypes in terms of recombination blocks. (The polymorphisms as such are determined through DNA sequencing of the parental strains.) Nevertheless, we agree with Reviewer #3 that this is conceptually close and that these papers established precedent. We therefore amended the manuscript and clearly state it in the Introduction (p.3). : "Further, although the sequence variation information contained in the RNA-seq data has been exploited for SNP detection (for instance: Quinn et al, 2013; Piskol et al, 2013), it has not been used in the framework of eQTL, possibly due to the complexity of the studied organisms (Keane et al, 2011; Montgomery et al, 2010, 2011; Pickrell et al, 2010)"

Concerning the expression quantification using strain-specific genomes: we agree that it has already been proposed in the framework of allele-specific expression, and we thank the reviewer for pointing out that the sentence "[t]his potential artifact in RNA-seq-based eQTL studies has not been investigated" was inappropriate. We therefore amended the manuscript to include in the main text references to the other approaches developed for allele specific expression (we include the references proposed by Reviewer #3 and others, and cited them in the Results and in the Discussion, pages 7 and 16-17, respectively). For instance on p.7: "However, such biases have already been noticed in the framework of allele specific expression in heterozygous diploid organisms (Degner et al, 2009; Franzén et al, 2013; Reddy et al, 2012; Satya et al, 2012; Turro et al, 2011; Stevenson et al, 2013; Pandey et al, 2013). As a solution to this problem it has been proposed to mask or exclude all or a part of the polymorphic positions in the reference genome prior to read-mapping (Degner et

al, 2009; Stevenson et al, 2013), or to used multi-mapping strategies (Turro et al, 2011; Satya et al, 2012)”. We also changed the erroneous sentence to: “The effects of this potential artifact whole-genome-RNA-seq-based eQTL mapping has not been investigated (to our knowledge), presumably because previous studies have used human samples where single-base genotype information is often not available (Montgomery et al, 2010; Pickrell et al, 2010; Majewski & Pastinen, 2011; Lalonde et al, 2011; Montgomery et al, 2011)” (page 7).

The authors that a large fraction of transcripts show linkage to trans-acting variants, a slightly unusual pattern in a data set like this. I would guess that it is a product of the serendipitous presence in their parent strain pair of two loci, an inversion on chromosome I and the swc5 locus they characterize (see below), that have very widespread effects on expression; it's not clear that this reveals any particular biological principle of relevance for the field.

This point has also been raised by Reviewer #1. We indeed agree that the particularity of the cross, in particular the chromosome 1 inversion and the *swc5* hotspot, can inflate the number of *trans*-eQTL. To test this, we repeated the *cis* versus *trans* analysis excluding the loci corresponding to the hotspots (including hotspot 11 corresponding to *swc5*-frame shift) and the entire chromosome I inversion. This indeed led to an increase of the *cis*-QTL proportion (6.8% of *cis*-eQTL). However this still remains much lower than what has been published and observed before (to our knowledge). This analysis has been added in the manuscript in the Discussion (p.15).

I was surprised to see that other than the sw5 locus, the authors say little about the biology of the mRNAs that vary between isolates or show linkage with markers, the Gene Ontology enrichment among mRNAs that show linkage to major trans-acting loci, their relationship to growth, etc. I acknowledge that it's been done in many systems already and that the antisense transcripts are what's new here, but it seems like an odd omission not to make some statements about the biological relevance of the bulk of their data in presenting Supplementary Table 2.

Indeed we did not exploit to the fullest the rich data presented here. We focused on the *swc5* hotspot and on antisense transcripts to have a clear focus for the publication. We actually think that this is a rich dataset that could yield many other biological insights regarding, for instance, splicing and the relation between macroscopic traits and molecular traits.

However we acknowledge the remark of Reviewer #3 and performed extra analyses of the eQTL hotspots in relation to the growth QTLs. In short, we found certain growth traits were linked to eQTL hotspots. A deeper analysis of these particular hotspots showed interesting results:

- The targets of hotspot 1 are enriched for genes involved in Ribosome Biogenesis, and hotspot 1 is linked to two growth traits (growth lag and area under the curve). Interestingly, links between ribosome biogenesis and growth are well established (via the Tor pathway).
- The targets of hotspot 3 are enriched for genes involved in stress response, and hotspot 3 is linked to a survival trait. Here also biological links between survival/longevity and stress response are well established.
- The targets of hotspot 5 are enriched for genes involved in energy metabolism. Moreover, hotspot 5 is also associated with the area under the growth curve. Once more, biological relationships between growth and energy metabolism are expected and have been shown before.

Altogether these results show the biological relevance of the data and of the strain library for studying growth, stress response and longevity, and to connect macroscopic traits with molecular traits.

These analyses have been added to manuscript in the Results section (paragraph about the hotspots, pages 9-10), and a statement about the biological interest of the data has been made (page 10)

Thank you again for submitting your work to Molecular Systems Biology. We have now heard back from the two referees who agreed to evaluate your manuscript. As you will see below, reviewer #2 thinks that their main concerns have been addressed. However, reviewer #1 still raises important concerns, which should be convincingly addressed in a revision of the manuscript.

In particular, reviewer #1 points out that additional analyses are required to more convincingly support the reported high trans-/cis-QTL ratio and to exclude that the number of trans-QTLs is inflated. As you may know, our editorial policy is to allow a single round of revision. While the issue raised by this reviewer is very important, it seems that it is potentially addressable. As such, we would ask you to convincingly address this point in an exceptional and last round of revision.

On a more editorial level, we would like to draw your attention to the following points:

- References to manuscripts in preparation should be cited in the text as (Author names(s), in preparation), and should not be included in the list of references.
- Please include the accession numbers and all other relevant information regarding the newly generated datasets (sequence data, RNA-Seq, ChIP-Seq) in the "Data Availability" section of your manuscript.

Please resubmit your revised manuscript online, with a covering letter listing amendments and responses to each point raised by the referees. Please resubmit the paper ****within one month**** and ideally as soon as possible. If we do not receive the revised manuscript within this time period, the file might be closed and any subsequent resubmission would be treated as a new manuscript. Please use the Manuscript Number (above) in all correspondence.

Reviewer #1:

The authors have addressed most of our previous suggestions. In budding yeast (as well as other organisms) previous results showed much more cis-QTLs compared to trans QTLs when the hotspots were removed. One of the major claims in this paper is the high ratio of trans/cis QTLs observed. However, with the release of the QTL mapping results, I noticed that the number of trans QTLs is indeed inflated (despite removal of chromosome 1 inversion and swc 5). This could not be assessed previously due to lack of data. Three possible reasons include

1. From dataset E11 (eQTL mapping results), many targets appear to be linked by multiple transQTLs in close proximity. For examples SPCC63.02c

SPCC63.02c chromosome_3 655654 663002 trans SPCC16C4.19 coding
 SPCC63.02c chromosome_3 664017 681957 trans SPCC16C4.06c coding
 SPCC63.02c chromosome_3 732326 738768 trans SPCC18B5.07c coding

the genotypes of these markers are highly correlated. It's not surprising that they would associate with the same target. Such case inflates the numbers of trans-QTLs reported. I suggest removing the correlated markers and collapsing them as one single QTL.

2. population structure is present in the experimental cross and this needs be accounted for in QTL mapping. Using the genotyping data (dataset E9), I observed population structure that is unexpected in the cross.

Plots

<http://docdroid.net/fz38>

<http://docdroid.net/fz3a>

At the intra-chosomal level, I observe marker linkage that doesn't agree with the physical distance. For example in chromosome 2, the markers on the telomeric end of the chromosome (~3.8Mb to 4Mb) are in high correlation (>0.7) with the marker MM378, chromosome2_2330408. Essentially if there is any true cis or trans eQTL in either of these regions, the other markers will be called as significant simply due to the high correlation. I also observed that there is substantially high correlation between markers of different chromosomes. I visualised the correlation of the genotype matrix using the plot.rf function from rqt package (see attached plot pairwise recombination and lod score). For example, there is a very large block in the telomeric end (~marker 380-480) of chromosome 2. This phenomenon is also observed at the inter-chosomal level. For example, a block around MM070 (1110692 to 1404507) on chromosome 1 has a correlation ~ 0.5 with MM440 in chromosome 2 (3505294 to 3507355). More cases can be found in the plot. It's

expected that the correlation drops close to zero when the markers are on different chromosomes. The presence of such high correlation between markers of different chromosomes suggests the population structure is quite strong in the cross. Without proper correction, the overall number of trans QTL would be inflated.

3. 10 kb cutoff for classifying cis and trans QTLs is rather arbitrary and too conservative. Such cutoff would falsely identify QTLs as trans when their effect is cis or they are in linkage with cis QTLs (related to point1) . By plotting the correlation (or R square) between markers as a function of distance, the correlation is still ~ 0.8 for markers that are 100 kb apart. This indicates that if a marker is 100kb away from the target gene, we cannot discount the fact that it could still be a cis QTL. Hence, depending on the marker correlation loosening the definition of cis marker to around 100kb or 200kb or even larger would make more sense.

the above points also apply to the aseQTLs results.

The population structure in this study is likely due to combination / compatibility of genes given the low viability of the cross. In normal QTL studies, the population structure (and correlation between markers) is corrected with mixed effect models (e.g EMMA). In this paper, a random forest regression that accounts for the population structure can be adapted for the QTL mapping.

Since the trans to cis QTL comparison is an integral part of the manuscript, I strongly recommend that the manuscript should be considered for publication only after the above concern is addressed.

Reviewer #2:

The manuscript of Clement-Ziza et al. has been extensively amended following the referees suggestions and comments.

The authors modified the figures (especially figure 5, 6 and 7) and have added novel analysis that fully clarify the critics. This manuscript warrants publication now in MSB journal without further modifications.

2nd Revision - authors' response

09 September 2014

We would like to thank the Reviewer #1, who thoughtfully analyzed our results and raised an important point. We have addressed all issues raised by the Reviewer through changing the QTL mapping and adapting our definition of cis-eQTLs (see eResponse to Reviewer1 for more details). We updated the manuscript, the figures (13 Figures modified), the tables, and the datasets. Although the remapping of QTLs indeed reduced the number of trans-eQTL, it did not alter our conclusions regarding the high trans/cis-QTL ratio. Moreover, none of the conclusions resulting from the reanalysis of the subsequent results, particularly regarding the swc5 QTL hotspot, were affected.

Reviewer #1:

The authors have addressed most of our previous suggestions. In budding yeast (as well as other organisms) previous results showed much more cis-QTLs compared to trans QTLs when the hotspots were removed. One of the major claims in this paper is the high ratio of trans/cis QTLs observed. However, with the release of the QTL mapping results, I noticed that the number of trans QTLs is indeed inflated (despite removal of chromosome 1 inversion and swc 5). This could not be assessed previously due to lack of data. Three possible reasons include

1. From dataset E11 (eQTL mapping results), many targets appear to be linked by multiple transQTLs in close proximity. For examples SPCC63.02c

*SPCC63.02c chromosome_3 655654 663002 trans SPCC16C4.19 coding
SPCC63.02c chromosome_3 664017 681957 trans SPCC16C4.06c coding
SPCC63.02c chromosome_3 732326 738768 trans SPCC18B5.07c coding*

the genotypes of these markers are highly correlated. It's not surprising that they would associate with the same target. Such case inflates the numbers of trans-QTLs reported. I suggest removing the correlated markers and collapsing them as one single QTL.

We agree with the Reviewer#1: traits linked to multiple correlated QTLs in close proximity can erroneously inflate the number of trans-QTLs, and should be collapsed. In order to address this point, we changed our definition of a QTL. Note that previously we already combined all contiguous markers linked to the same trait into a single locus (QTL). However, if such markers were separated by one or more non-linked markers, we treated them as distinct regions (i.e. several QTLs). We changed this scheme: in the revised version we combined such QTL 'regions' whenever markers in the two (or more) regions were in linkage disequilibrium (LD, estimated via the Pearson product-moment correlation coefficient $r > 0.8$ between the most correlated markers of the linked region), and the linked regions were not separated by more than ten markers. All of those markers (including the intermediate non-linked markers) were combined into one QTL (one region). The method is described in the Methods section of the manuscript (p.31).

Please see below (response to subsequent points) for how we dealt with cases when such regions were separated by more than 10 markers.

Applying this rule, the example highlighted by the Reviewer will be collapsed into a single QTL on chromosome_3 ranging from the position 655654 to 738768. In the revised version of the Dataset E11 such cases or similar are not anymore present.

2. population structure is present in the experimental cross and this needs be accounted for in QTL mapping. Using the genotyping data (dataset E9), I observed population structure that is unexpected in the cross.

Plots

<http://docdroid.net/fz38>

<http://docdroid.net/fz3a>

At the intra-chosomal level, I observe marker linkage that doesn't agree with the physical distance. For example in chromosome 2, the markers on the telomeric end of the chromosome (~3.8Mb to 4Mb) are in high correlation (>0.7) with the marker MM378, chromosome2_2330408. Essentially if there is any true cis or trans eQTL in either of these regions, the other markers will be called as significant simply due to the high correlation. I also observed that there is substantially high correlation between markers of different chromosomes. I visualised the correlation of the genotype matrix using the plot.rf function from rqt package (see attached plot pairwise recombination and lod score). For example, there is a very large block in the telomeric end (~marker 380-480) of chromosome 2. This phenomenon is also observed at the inter-chosomal level. For example, a block around MM070 (1110692 to 1404507) on chromosome 1 has a correlation ~ 0.5 with MM440 in chromosome 2 (3505294 to 3507355). More cases can be found in the plot. It's expected that the correlation drops close to zero when the markers are on different chromosomes. The presence of such high correlation between markers of different chromosomes suggests the population structure is quite strong in the cross. Without proper correction, the overall number of trans QTL would be inflated.

The population structure in this study is likely due to combination / compatibility of genes given the low viability of the cross. In normal QTL studies, the population structure (and correlation between markers) is corrected with mixed effect models (e.g EMMA). In this paper, a random forest regression that accounts for the population structure can be adapted for the QTL mapping.

We thank Reviewer #1 for thoughtfully pointing out potential problems with population structure in the cross. As she/he suggested we remapped the eQTL and the aseQTL while accounting for potential population structure. We adapted previously proposed methods (Price et al, 2006; Novembre & Stephens, 2008; Price et al, 2010; Patterson et al, 2006; see manuscript references for details) that integrate the population structure as covariates in the QTL model to our Random Forest approach. First we estimated the kinship matrix using the *emma* package and used the top 8 eigenvectors as additional predictors in the QTL mapping (Random Forest growing). Here the first 8 vectors explained more than 80% of the genotype variance. We took care that the permutations used to estimate the significance of the linkages maintained the correspondence between the covariates

(population structure models) and the permuted traits. The method is fully described in the Methods section (p.30).

Although virtually all numbers reported in the manuscript changed a bit, none of our conclusions was affected. Explicitly modelling relatedness did not substantially change our conclusions also due to the fact that the standard Random Forest approach already partly corrects for population structure. Multi marker mapping methods (that consider all markers together instead of one-by-one) are less affected by population sub-structure than single marker mapping methods, because markers ‘just’ explaining the genetic background are less likely to be selected. Moreover, bootstrap aggregation methods (bagging)—Random Forest is one—have been shown to intrinsically correct, to some extent, for population structure (Vladar *et al*, 2009, PMID: 19474203).

To further account for the correlation between distant markers due to population structure, we grouped QTLs in high LD. In addition to collapsing QTLs in LD in close proximity, we grouped QTLs in LD (estimated as previously via the Pearson product-moment correlation coefficient $r > 0.8$ between the most correlated markers of the QTLs) when they were located on different chromosomes or separated by more than 10 markers (Methods section p.30). We refer to these sets of markers as ‘QTL groups’, because they do not constitute a contiguous genomic region (i.e. not a single ‘locus’). QTL groups were counted as single QTL in the subsequent analyses (in particular for *cis* versus *trans*).

3. 10 kb cutoff for classifying cis and trans QTLs is rather arbitrary and too conservative. Such cutoff would falsely identify QTLs as trans when their effect is cis or they are in linkage with cis QTLs (related to point1) . By plotting the correlation (or R square) between markers as a function of distance, the correlation is still ~ 0.8 for markers that are 100 kb apart. This indicates that if a marker is 100kb away from the target gene, we cannot discount the fact that it could still be a cis QTL. Hence, depending on the marker correlation loosening the definition of cis marker to around 100kb or 200kb or even larger would make more sense.

Instead of using a physical distance threshold between the QTL and the target genes to distinguish *cis* from *trans* effects, we propose in this revised version of the manuscript to use correlation thresholds to define *cis*-eQTL, which better reflects genetic distance. This approach has the advantage to take into account the issue pointed out by the Reviewer in a marker specific manner. For each QTL (or QTL group), we compute the Spearman correlation between the markers defining the QTLs and the markers surrounding the target genes. When the correlation is above 0.8 ($r > 0.8$), the QTL is considered acting in *cis* regardless of its genomic distance to the target gene (see Methods p.31).

To further confirm that the high proportion of *trans*-eQTLs that we observed was not due to a too conservative definition of the *cis* effects, we relaxed the threshold to $r > 0.7$ and $r > 0.6$, which did not change our conclusions (see Discussion p.15).

the above points also apply to the aseQTLs results.

The aseQTL and the eQTL were remapped applying the same revised methods in this version of the manuscript.

Since the trans to cis QTL comparison is an integral part of the manuscript, I strongly recommend that the manuscript should be considered for publication only after the above concern is addressed.

In this revised version of the manuscript, (i) we explicitly corrected for potential population structure in the eQTL and the aseQTL mapping, (ii) we modified the definition of the QTL intervals to collapse correlated QTLs located in close distance into single locus, (iii) we introduced QTL groups to further take into account the problem of distant correlated markers, and (iv) we changed the definition of the *cis* effect. Although these changes lead to a reduced numbers of *trans*-eQTLs (suggesting that the previously reported numbers of *trans*-eQTLs were indeed inflated, as pointed by the Reviewer #1), the proportion of *trans*-eQTLs (89.6%) remained much higher than in previous studies. Therefore, these changes did not modify our previous conclusions.

Since the QTL mapping was one of the first steps of our analysis pipeline, we reanalysed, retested and reconsidered all the results derived from the QTL mapping (in particular the analyses of the *swc5* hotspot). All the resulting Figures (Figures 3, 5, 6, 8, E17, E18, E19, E20, E24, E25, E26, E27, E32), tables (Tables E3, E4, E5), datasets (datasets E10, E11, E12, E13, E15), and analyses (manuscript and Extended View Text) were revised. Altogether, this did not change our conclusions and the messages of the manuscript.

Thank you again for submitting your work to Molecular Systems Biology. We have now heard back from the referee who agreed to evaluate your manuscript. As you will see below, reviewer #1 is satisfied with the modifications made and supports publication of the work. However, the reviewer refers to the need to include some relatively minor modifications, which we would ask you to address in a revision of this work.

Reviewer #1:

The authors have sufficiently addressed my suggestions and concerns in the latest revision. Authors may want to reflect the presence of the population structure in their cross in the manuscript, which would help readers to understand potential issues in the QTL mapping and also for mapping other traits using the same cross.

Two minor things,

1. The start and end positions of many QTL in both dataset E11 (list of the significant eQTL with their genomic position, target and putative regulator) and dataset E13 (list of the significant aseQTL with their position and target) are erroneously reported. They don't correspond to the correct QTL reported in Dataset E10 and E12 respectively. Many cannot be found in the marker information table (Dataset E9).

2. Typo on page 8. "Four out of height eQTL hotspots were also aseQTL hotspots". It should be four out of eight. +

I'd recommend the publication of the manuscript in MSB journal.

Reviewer #1:

The authors have sufficiently addressed my suggestions and concerns in the latest revision. Authors may want to reflect the presence of the population structure in their cross in the manuscript, which would help readers to understand potential issues in the QTL mapping and also for mapping other traits using the same cross.

As suggested by Reviewer #1, we amended the manuscript and added this sentence: "Because we noticed the presence of population structure in the cross, The QTL mapping method was improved to also account for population structure and missing genotype data" in the result section. (p.7)

Two minor things,

1. The start and end positions of many QTL in both dataset E11 (list of the significant eQTL with their genomic position, target and putative regulator) and dataset E13 (list of the significant aseQTL with their position and target) are erroneously reported. They don't correspond to the correct QTL reported in Dataset E10 and E12 respectively. Many cannot be found in the marker information table (Dataset E9).

We thank Reviewer #1 for noticing these discrepancies that made us realize that we did not describe well enough how these table were generated. Indeed, QTLs were considered as spanning in between the firsts genomic variants not linked to the trait. That thus correspond to the "end" position of the previous not linked marker to the "start" position of the next not linked marker (as referred to Supplementary Dataset S9). We amended the manuscript in the method section p. 31 and in the descriptions of the Supplementary Dataset S11 and S13 to clarify this point. Beside, some other mistakes in the reporting of the end coordinates in the Supplementary Datasets S11 and S13 were also corrected.

2. Typo on page 8. "Four out of height eQTL hotspots were also aseQTL hotspots". It should be four out of eight. +. This has been corrected.